# Understanding Diffusion Objectives as the ELBO with Simple Data Augmentation

**Diederik P. Kingma**
Google DeepMind
durk@google.com

**Ruiqi Gao**
Google DeepMind
ruiqig@google.com

## Abstract

To achieve the highest perceptual quality, state-of-the-art diffusion models are optimized with objectives that typically look very different from the maximum likelihood and the Evidence Lower Bound (ELBO) objectives. In this work, we reveal that diffusion model objectives are actually closely related to the ELBO.

Specifically, we show that all commonly used diffusion model objectives equate to a weighted integral of ELBOs over different noise levels, where the weighting depends on the specific objective used. Under the condition of monotonic weighting, the connection is even closer: the diffusion objective then equals the ELBO, combined with simple data augmentation, namely Gaussian noise perturbation. We show that this condition holds for a number of state-of-the-art diffusion models.

In experiments, we explore new monotonic weightings and demonstrate their effectiveness, achieving state-of-the-art FID scores on the high-resolution ImageNet benchmark.

## 1 Introduction

Diffusion-based generative models, or diffusion models in short, were first introduced by Sohl-Dickstein et al. [2015]. After years of relative obscurity, this class of models suddenly rose to prominence with the work of Song and Ermon [2019] and Ho et al. [2020] who demonstrated that, with further refinements in model architectures and objective functions, diffusion models can perform state-of-the-art image generation.

37th Conference on Neural Information Processing Systems (NeurIPS 2023).

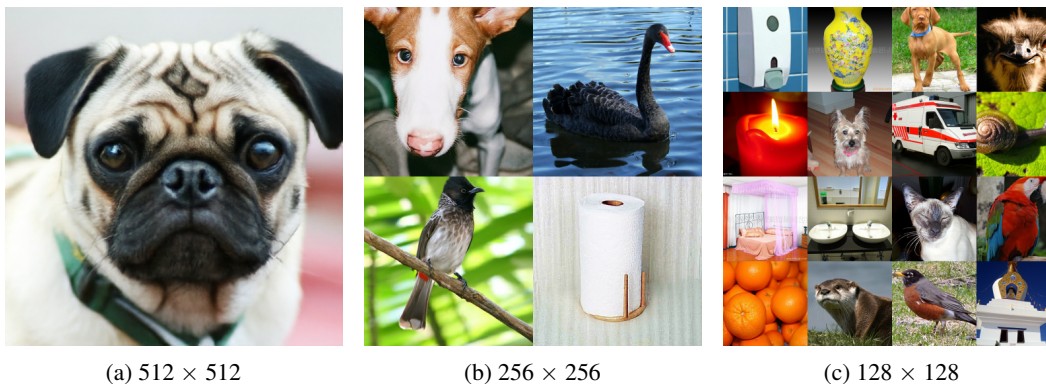

(a) $512 \times 512$  (b) $256 \times 256$  (c) $128 \times 128$

Figure 1: Samples generated from our *VDM++* diffusion models trained on the ImageNet dataset; see Section 5 for details and Appendix M for more samples.

The diffusion model framework has since been successfully applied to text-to-image generation [Rombach et al., 2022b, Nichol et al., 2021, Ramesh et al., 2022, Saharia et al., 2022b, Yu et al., 2022, Nichol and Dhariwal, 2021, Ho et al., 2022, Dhariwal and Nichol, 2022, Ding et al., 2021], image-to-image generation [Saharia et al., 2022c,a, Whang et al., 2022], 3D synthesis [Poole et al., 2022, Watson et al., 2022], text-to-speech [Chen et al., 2021a, Kong et al., 2021, Chen et al., 2021b], and density estimation [Kingma et al., 2021, Song et al., 2021a].

Diffusion models can be interpreted as a special case of deep variational autoencoders (VAEs) [Kingma and Welling, 2013, Rezende et al., 2014] with a particular choice of inference model and generative model. Just like VAEs, the original diffusion models [Sohl-Dickstein et al., 2015] were optimized by maximizing the variational lower bound of the log-likelihood of the data, also called the evidence lower bound, or ELBO for short. It was shown by *Variational Diffusion Models* (VDM) [Kingma et al., 2021] and [Song et al., 2021a] how to optimize *continuous-time* diffusion models with the ELBO objective, achieving state-of-the-art likelihoods on image density estimation benchmarks.

However, the best results in terms of sample quality metrics such as FID scores were achieved with other objectives, for example a denoising score matching objective [Song and Ermon, 2019] or a simple noise-prediction objective [Ho et al., 2020]. These now-popular objective functions look, on the face of it, very different from the traditionally popular maximum likelihood and ELBO objectives. Through the analysis in this paper, we reveal that all training objectives used in state-of-the-art diffusion models are actually closely related to the ELBO objective.

This paper is structured as follows:

- In Section 2 we introduce the broad diffusion model family under consideration.
- In Section 3, we show how the various diffusion model objectives in the literature can be understood as special cases of a *weighted loss* [Kingma et al., 2021, Song et al., 2021a], with different choices of weighting. The weighting function specifies the weight per noise level. In Section 3.2 we show that during training, the noise schedule acts as a importance sampling distribution for estimating the loss, and is thus important for efficient optimization. Based on this insight we propose a simple adaptive noise schedule.
- In Section 4, we present our main result: that if the weighting function is a monotonic function of time, then the weighted loss corresponds to maximizing the ELBO with data augmentation, namely Gaussian noise perturbation. This holds for, for example, the **v**-prediction loss of [Salimans and Ho, 2022] and flow matching with the optimal transport path [Lipman et al., 2022].
- In Section 5 we perform experiments with various new monotonic weights on the ImageNet dataset, and find that our proposed monotonic weighting produces models with sample quality that are competitive with the best published results, achieving state-of-art FID and IS scores on high resolution ImageNet generation.

## 1.1 Related work

The main sections reference much of the related work. Earlier work [Kingma et al., 2021, Song et al., 2021a, Huang et al., 2021, Vahdat et al., 2021], including Variational Diffusion Models [Kingma et al., 2021], showed how to optimize continous-time diffusion models towards the ELBO objective. We generalize these earlier results by showing that any diffusion objective that corresponds with monotonic weighting corresponds to the ELBO, combined with a form of DistAug [Child et al., 2019]. DistAug is a method of training data distribution augmentation for generative models where the model is conditioned on the data augmentation parameter at training time, and conditioned on 'no augmentation' at inference time. The type of data augmentation under consideration in this paper, namely additive Gaussian noise, is also a form of data distribution smoothing, which has been shown to improve sample quality in autoregressive models by Meng et al. [2021].

Kingma et al. [2021] showed how the ELBO is invariant to the choice of noise schedule, except for the endpoints. We generalize this result by showing that the invariance holds for any weighting function.

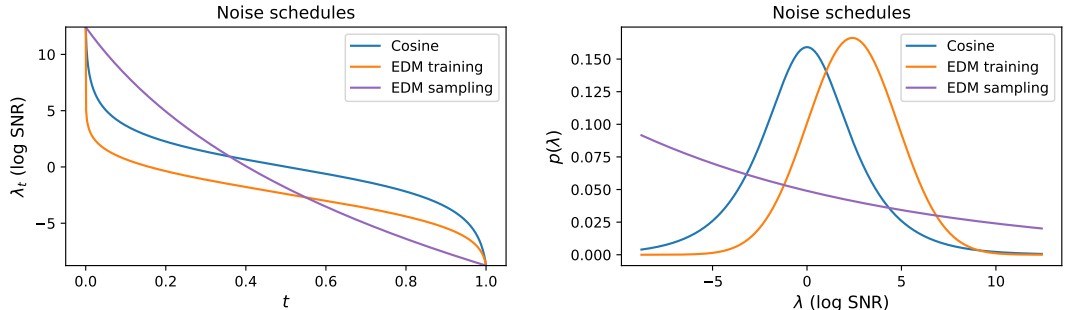

Figure 2: **Left:** Noise schedules used in our experiments: cosine [Nichol and Dhariwal, 2021] and EDM [Karras et al., 2022] training and sampling schedules. **Right:** The same noise schedules, expressed as probability densities $p(\lambda) = -dt/d\lambda$. See Section 2.1 and Appendix E.3 for details.

## 2 Model

Suppose we have a dataset of datapoints drawn from $q(\mathbf{x})$. We wish to learn a generative model $p_{\boldsymbol{\theta}}(\mathbf{x})$ that approximates $q(\mathbf{x})$. We'll use shorthand notation $p := p_{\boldsymbol{\theta}}$.

The observed variable $\mathbf{x}$ might be the output of a pre-trained encoder, as in *latent diffusion models* [Vahdat et al., 2021, Rombach et al., 2022a], on which the popular *Stable Diffusion* model is based. Our theoretical analysis also applies to this type of model.

In addition to the observed variable $\mathbf{x}$, we have a series of latent variables $\mathbf{z}_t$ for timesteps $t \in [0, 1]$: $\mathbf{z}_{0,\dots,1} := \mathbf{z}_0, \dots, \mathbf{z}_1$. The model consists of two parts: a *forward process* forming a conditional joint distribution $q(\mathbf{z}_{0,\dots,1}|\mathbf{x})$, and a *generative model* forming a joint distribution $p(\mathbf{z}_{0,\dots,1})$.

### 2.1 Forward process and noise schedule

The forward process is a Gaussian diffusion process, giving rise to a conditional distribution $q(\mathbf{z}_{0,\dots,1}|\mathbf{x})$; see Appendix E.1 for details. For every $t \in [0, 1]$, the marginal distribution $q(\mathbf{z}_t|\mathbf{x})$ is given by:

$$\mathbf{z}_t = \alpha_\lambda \mathbf{x} + \sigma_\lambda \boldsymbol{\epsilon} \quad \text{where} \quad \boldsymbol{\epsilon} \sim \mathcal{N}(0, \mathbf{I}). \tag{1}$$

In case of the often-used *variance preserving* (VP) forward process, $\alpha_\lambda^2 = \text{sigmoid}(\lambda_t)$ and $\sigma_\lambda^2 = \text{sigmoid}(-\lambda_t)$, but other choices are possible; our results are agnostic to this choice. The *log signal-to-noise ratio* (log-SNR) for timestep $t$ is given by $\lambda = \log(\alpha_\lambda^2/\sigma_\lambda^2)$.

The *noise schedule* is a strictly monotonically decreasing function $f_\lambda$ that maps from the time variable $t \in [0, 1]$ to the corresponding log-SNR $\lambda$: $\lambda = f_\lambda(t)$. We sometimes denote the log-SNR as $\lambda_t$ to emphasize that it is a function of $t$. The endpoints of the noise schedule are given by $\lambda_{\max} := f_\lambda(0)$ and $\lambda_{\min} := f_\lambda(1)$. See Figure 2 for a visualization of commonly used noise schedules in the literature, and Appendix E.3 for more details.

Due to its monotonicity, $f_\lambda$ is invertible: $t = f_\lambda^{-1}(\lambda)$. Given this bijection, we can do a change of variables: a function of the value $t$ can be equivalently written as a function of the corresponding value $\lambda$, and vice versa, which we'll make use of in this work.

During model training, we sample time $t$ uniformly: $t \sim \mathcal{U}(0, 1)$, then compute $\lambda = f_\lambda(t)$. This results in a distribution over noise levels $p(\lambda) = -dt/d\lambda = -1/f_\lambda'(t)$ (see Section E.3), which we also plot in Figure 2.

Sometimes it is best to use a different noise schedule for sampling from the model than for training. During sampling, the density $p(\lambda)$ gives the relative amount of time the sampler spends at different noise levels.

## 2.2 Generative model

The data $\mathbf{x} \sim \mathcal{D}$, with density $q(\mathbf{x})$, plus the forward model defines a joint distribution $q(\mathbf{z}_0, ..., \mathbf{z}_1) = \int q(\mathbf{z}_0, ..., \mathbf{z}_1 | \mathbf{x}) q(\mathbf{x}) d\mathbf{x}$, with marginals $q_t(\mathbf{z}) := q(\mathbf{z}_t)$. The generative model defines a corresponding joint distribution over latent variables: $p(\mathbf{z}_0, ..., \mathbf{z}_1)$.

For large enough $\lambda_{\max}$, $\mathbf{z}_0$ is almost identical to $\mathbf{x}$, so learning a model $p(\mathbf{z}_0)$ is practically equivalent to learning a model $p(\mathbf{x})$. For small enough $\lambda_{\min}$, $\mathbf{z}_1$ holds almost no information about $\mathbf{x}$, such that there exists a distribution $p(\mathbf{z}_1)$ satisfying $D_{KL}(q(\mathbf{z}_1|\mathbf{x})||p(\mathbf{z}_1)) \approx 0$. Usually we can use $p(\mathbf{z}_1) = \mathcal{N}(0, \mathbf{I})$.

Let $\mathbf{s}_{\boldsymbol{\theta}}(\mathbf{z}; \lambda)$ denote a *score model*, which is a neural network that we let approximate $\nabla_{\mathbf{z}} \log q_t(\mathbf{z})$ through methods introduced in the next sections. If $\mathbf{s}_{\boldsymbol{\theta}}(\mathbf{z}; \lambda) = \nabla_{\mathbf{z}} \log q_t(\mathbf{z})$, then the forward process can be exactly reversed; see Appendix E.4.

If $D_{KL}(q(\mathbf{z}_1)||p(\mathbf{z}_1)) \approx 0$ and $\mathbf{s}_{\boldsymbol{\theta}}(\mathbf{z}; \lambda) \approx \nabla_{\mathbf{z}} \log q_t(\mathbf{z})$, then we have a good generative model in the sense that $D_{KL}(q(\mathbf{z}_{0,...,1})||p(\mathbf{z}_{0,...,1})) \approx 0$, which implies that $D_{KL}(q(\mathbf{z}_0)||p(\mathbf{z}_0)) \approx 0$ which achieves our goal. So, our generative modeling task is reduced to learning a score network $\mathbf{s}_{\boldsymbol{\theta}}(\mathbf{z}; \lambda)$ that approximates $\nabla_{\mathbf{z}} \log q_t(\mathbf{z})$.

Sampling from the generative model can be performed by sampling $\mathbf{z}_1 \sim p(\mathbf{z}_1)$, then (approximately) solving the reverse SDE using the estimated $\mathbf{s}_{\boldsymbol{\theta}}(\mathbf{z}; \lambda)$. Recent diffusion models have used increasingly sophisticated procedures for approximating the reverse SDE; see Appendix E.4. In experiments we use the DDPM sampler from Ho et al. [2020] and the stochastic sampler with Heun's second order method proposed by Karras et al. [2022].

# 3 Diffusion Model Objectives

**Denoising score matching.** Above, we saw that we need to learn a score network $\mathbf{s}_{\boldsymbol{\theta}}(\mathbf{z}; \lambda_t)$ that approximates $\nabla_{\mathbf{z}} \log q_t(\mathbf{z})$, for all noise levels $\lambda_t$. It was shown by [Vincent, 2011, Song and Ermon, 2019] that this can be achieved by minimizing a denoising score matching objective over all noise scales and all datapoints $\mathbf{x} \sim \mathcal{D}$:

$$\mathcal{L}_{\text{DSM}}(\mathbf{x}) = \mathbb{E}_{t \sim \mathcal{U}(0,1), \boldsymbol{\epsilon} \sim \mathcal{N}(0,\mathbf{I})} \left[ \tilde{w}(t) \cdot ||\mathbf{s}_{\boldsymbol{\theta}}(\mathbf{z}_t; \lambda_t) - \nabla_{\mathbf{z}_t} \log q(\mathbf{z}_t|\mathbf{x})||_2^2 \right] \tag{2}$$

where $\mathbf{z}_t = \alpha_\lambda \mathbf{x} + \sigma_\lambda \boldsymbol{\epsilon}$.

**The $\epsilon$-prediction objective.** Most contemporary diffusion models are optimized towards a noise-prediction loss introduced by [Ho et al., 2020]. In this case, the score network is typically parameterized through a noise-prediction ($\epsilon$-prediction) model: $\mathbf{s}_{\boldsymbol{\theta}}(\mathbf{z}; \lambda) = -\hat{\boldsymbol{\epsilon}}_{\boldsymbol{\theta}}(\mathbf{z}; \lambda)/\sigma_\lambda$. (Other options, such as $\mathbf{x}$-prediction, $\mathbf{v}$-prediction, and EDM parameterizations, are explained in Appendix E.2.) The noise-prediction loss is:

$$\mathcal{L}_{\boldsymbol{\epsilon}}(\mathbf{x}) = \frac{1}{2} \mathbb{E}_{t \sim \mathcal{U}(0,1), \boldsymbol{\epsilon} \sim \mathcal{N}(0,\mathbf{I})} \left[ ||\hat{\boldsymbol{\epsilon}}_{\boldsymbol{\theta}}(\mathbf{z}_t; \lambda_t) - \boldsymbol{\epsilon}||_2^2 \right] \tag{3}$$

Since $||\mathbf{s}_{\boldsymbol{\theta}}(\mathbf{z}_t; \lambda_t) - \nabla_{\mathbf{x}_t} \log q(\mathbf{z}_t|\mathbf{x})||_2^2 = \sigma_\lambda^{-2} ||\hat{\boldsymbol{\epsilon}}_{\boldsymbol{\theta}}(\mathbf{z}_t; \lambda_t) - \boldsymbol{\epsilon}||_2^2$, this is simply a version of the denoising score matching objective in Equation 2, but where $\tilde{w}(t) = \sigma_t^2$. Ho et al. [2020] showed that this noise-prediction objective can result in high-quality samples. Dhariwal and Nichol [2022] later improved upon these results by switching from a 'linear' to a 'cosine' noise schedule $\lambda_t$ (see Figure 2). This noise-prediction loss with the cosine schedule is currently broadly used.

**The ELBO objective.** It was shown by [Kingma et al., 2021, Song et al., 2021a] that the evidence lower bound (ELBO) of continuous-time diffusion models simplifies to:

$$-\text{ELBO}(\mathbf{x}) = \frac{1}{2} \mathbb{E}_{t \sim \mathcal{U}(0,1), \boldsymbol{\epsilon} \sim \mathcal{N}(0,\mathbf{I})} \left[ -\frac{d\lambda}{dt} \cdot ||\hat{\boldsymbol{\epsilon}}_{\boldsymbol{\theta}}(\mathbf{z}_t; \lambda_t) - \boldsymbol{\epsilon}||_2^2 \right] + c \tag{4}$$

where $c$ is constant w.r.t. the score network parameters.

| Loss function | Implied weighting $w(\lambda)$ | Monotonic? |
|---|---|---|
| ELBO [Kingma et al., 2021, Song et al., 2021a] | 1 | ✓ |
| IDDPM ($\epsilon$-prediction with 'cosine' schedule) [Nichol and Dhariwal, 2021] | $\mathrm{sech}(\lambda/2)$ | |
| EDM [Karras et al., 2022] (Appendix D.1) | $\mathcal{N}(\lambda; 2.4, 2.4^2) \cdot (e^{-\lambda} + 0.5^2)$ | |
| $\mathbf{v}$-prediction with 'cosine' schedule [Salimans and Ho, 2022] (Appendix D.2) | $e^{-\lambda/2}$ | ✓ |
| Flow Matching with OT path (FM-OT) [Lipman et al., 2022] (Appendix D.3) | $e^{-\lambda/2}$ | ✓ |
| InDI [Delbracio and Milanfar, 2023] (Appendix D.4) | $e^{-\lambda}\mathrm{sech}^2(\lambda/4)$ | ✓ |
| P2 weighting with 'cosine' schedule [Choi et al., 2022] (Appendix D.5) | $\mathrm{sech}(\lambda/2)/(1 + e^\lambda)^\gamma, \gamma = 0.5$ or 1 | |
| Min-SNR-$\gamma$ [Hang et al., 2023] (Appendix D.6) | $\mathrm{sech}(\lambda/2) \cdot \min(1, \gamma e^{-\lambda})$ | |

Table 1: Diffusion model objectives in the literature are special cases of the weighted loss with a weighting function $w(\lambda)$ given in this table. See Section 3.1 and Appendix D for more details and derivations. Most existing weighting functions are non-monotonic, except for the ELBO objective and the $\mathbf{v}$-prediction objective with 'cosine' schedule.

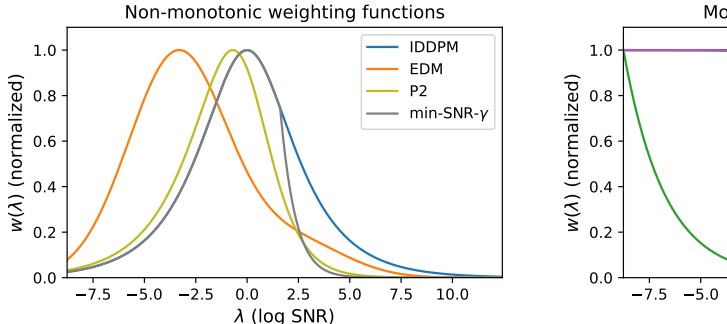

Figure 3: Diffusion model objectives in the literature are special cases of the weighted loss with non-monotonic (left) or monotonic (right) weighting functions. Each weighting function is scaled such that the maximum is 1 over the plotted range. See Table 1 and Appendix D.

## 3.1 The weighted loss

The different objective functions used in practice, including the ones above, can be viewed as special cases of the *weighted loss* introduced by Kingma et al. [2021][1] with a different choices of weighting function $w(\lambda_t)$:

$$\mathcal{L}_w(\mathbf{x}) = \frac{1}{2}\mathbb{E}_{t \sim \mathcal{U}(0,1), \epsilon \sim \mathcal{N}(0,\mathbf{I})}\left[w(\lambda_t) \cdot -\frac{d\lambda}{dt} \cdot ||\hat{\boldsymbol{\epsilon}}_{\boldsymbol{\theta}}(\mathbf{z}_t; \lambda_t) - \boldsymbol{\epsilon}||_2^2\right] \qquad (5)$$

See Appendix D for a derivation of the implied weighting functions for all popular diffusion losses. Results are compiled in Table 1, and visualized in Figure 3.

The ELBO objective (Equation 4) corresponds to uniform weighting, i.e. $w(\lambda_t) = 1$.

The popular noise-prediction objective (Equation 3) corresponds to $w(\lambda_t) = -1/(d\lambda/dt)$. This is more compactly expressed as $w(\lambda_t) = p(\lambda_t)$, i.e., the PDF of the implied distribution over noise levels $\lambda$ during training. Typically, the noise-prediction objective is used with the cosine schedule $\lambda_t$, which implies $w(\lambda_t) \propto \mathrm{sech}(\lambda_t/2)$. See Section E.3 for the expression of $p(\lambda_t)$ for various noise schedules.

## 3.2 Invariance of the weighted loss to the noise schedule $\lambda_t$

In Kingma et al. [2021], it was shown that the ELBO objective (Equation 4) is invariant to the choice of noise schedule, except for its endpoints $\lambda_{\min}$ and $\lambda_{\max}$. This result can be extended to the much

---

[1]More specifically, Kingma et al. [2021] expressed the weighted diffusion loss in terms of $\mathbf{x}$-prediction, which is equivalent to the expression above due to the relationship $\int ||\boldsymbol{\epsilon} - \hat{\boldsymbol{\epsilon}}_{\boldsymbol{\theta}}(\mathbf{z}_t; \lambda)||_2^2 d\lambda = \int ||\mathbf{x} - \hat{\mathbf{x}}_{\boldsymbol{\theta}}(\mathbf{z}_t; \lambda)||_2^2 e^\lambda d\lambda = \int ||\mathbf{x} - \hat{\mathbf{x}}_{\boldsymbol{\theta}}(\mathbf{z}_t; \lambda)||_2^2 de^\lambda$, where $e^\lambda$ equals the signal-to-noise ratio (SNR).

more general weighted diffusion loss of Equation 5, since with a change of variables from $t$ to $\lambda$, it can be rewritten to:

$$\mathcal{L}_w(\mathbf{x}) = \frac{1}{2} \int_{\lambda_{\min}}^{\lambda_{\max}} w(\lambda) \mathbb{E}_{\boldsymbol{\epsilon} \sim \mathcal{N}(0, \mathbf{I})} \left[ ||\hat{\boldsymbol{\epsilon}}_{\boldsymbol{\theta}}(\mathbf{z}_\lambda; \lambda) - \boldsymbol{\epsilon}||_2^2 \right] d\lambda \tag{6}$$

Note that this integral does not depend on the noise schedule $f_\lambda$ (the mapping between $t$ and $\lambda$), except for its endpoints $\lambda_{\max}$ and $\lambda_{\min}$. The shape of the function $f_\lambda$ between $\lambda_{\min}$ and $\lambda_{\max}$ does not affect the loss; only the weighting function $w(\lambda)$ does. Given a chosen weighting function $w(\lambda)$, the loss is invariant to the noise schedule $\lambda_t$ between $t = 0$ and $t = 1$. This is important, since it means that the only real difference between diffusion objectives is their difference in weighting $w(\lambda)$.

This invariance does *not* hold for the Monte Carlo estimator of the loss that we use in training, based on random samples $t \sim \mathcal{U}(0, 1)$, $\boldsymbol{\epsilon} \sim \mathcal{N}(0, \mathbf{I})$. The noise schedule still affects the *variance* of this Monte Carlo estimator and its gradients; therefore, the noise schedule affects the efficiency of optimization. More specifically, we'll show that the noise schedule acts as an importance sampling distribution for estimating the loss integral of Equation 6. Note that $p(\lambda) = -1/(d\lambda/dt)$. We can therefore rewrite the weighted loss as the following, which clarifies the role of $p(\lambda)$ as an importance sampling distribution:

$$\mathcal{L}_w(\mathbf{x}) = \frac{1}{2} \mathbb{E}_{\boldsymbol{\epsilon} \sim \mathcal{N}(0, \mathbf{I}), \lambda \sim p(\lambda)} \left[ \frac{w(\lambda)}{p(\lambda)} ||\hat{\boldsymbol{\epsilon}}_{\boldsymbol{\theta}}(\mathbf{z}_\lambda; \lambda) - \boldsymbol{\epsilon}||_2^2 \right] \tag{7}$$

Using this insight, we implemented an adaptive noise schedule, detailed in Appendix F. We find that by lowering the variance of the loss estimator, this often significantly speeds op optimization.

## 4    The weighted loss as the ELBO with data augmentation

We prove the following main result:

> **Theorem 1.** *If the weighting $w(\lambda_t)$ is monotonic, then the weighted diffusion objective of Equation 5 is equivalent to the ELBO with data augmentation (additive noise).*

With monotonic $w(\lambda_t)$ we mean that $w$ is a monotonically increasing function of $t$, and therefore a monotonically decreasing function of $\lambda$.

We'll use shorthand notation $\mathcal{L}(t; \mathbf{x})$ for the KL divergence between the joint distributions of the forward process $q(\mathbf{z}_{t,...1}|\mathbf{x})$ and the reverse model $p(\mathbf{z}_{t,...1})$, for the subset of timesteps from $t$ to 1:

$$\mathcal{L}(t; \mathbf{x}) := D_{KL}(q(\mathbf{z}_{t,...,1}|\mathbf{x})||p(\mathbf{z}_{t,...,1})) \tag{8}$$

In Appendix A.1, we prove that[2]:

$$\frac{d}{dt}\mathcal{L}(t; \mathbf{x}) = \frac{1}{2} \frac{d\lambda}{dt} \mathbb{E}_{\boldsymbol{\epsilon} \sim \mathcal{N}(0, \mathbf{I})} \left[ ||\boldsymbol{\epsilon} - \hat{\boldsymbol{\epsilon}}_{\boldsymbol{\theta}}(\mathbf{z}_\lambda; \lambda)||_2^2 \right] \tag{9}$$

As shown in Appendix A.1, this allows us to rewrite the weighted loss of Equation 5 as simply:

$$\mathcal{L}_w(\mathbf{x}) = -\int_0^1 \frac{d}{dt}\mathcal{L}(t; \mathbf{x}) \, w(\lambda_t) \, dt \tag{10}$$

In Appendix A.2, we prove that using integration by parts, the weighted loss can then be rewritten as:

$$\mathcal{L}_w(\mathbf{x}) = \int_0^1 \frac{d}{dt} w(\lambda_t) \, \mathcal{L}(t; \mathbf{x}) \, dt + w(\lambda_{\max}) \, \mathcal{L}(0; \mathbf{x}) + \text{constant} \tag{11}$$

---

[2]Interestingly, this reveals a new relationship between the KL and Fisher divergences: $\frac{d}{d\lambda} D_{KL}(q(\mathbf{z}_{t,...,1}|\mathbf{x})||p(\mathbf{z}_{t,...,1})) = \frac{1}{2}\sigma_\lambda^2 D_F(q(\mathbf{z}_t|\mathbf{x})||p(\mathbf{z}_t))$. See Appendix G for a derivation, a discussion, and a comparison with a similar result by Lyu [2012].

Now, assume that $w(\lambda_t)$ is a monotonically increasing function of $t \in [0,1]$. Also, without loss of generality, assume that $w(\lambda_t)$ is normalized such that $w(\lambda_1) = 1$. We can then further simplify the weighted loss to an expected KL divergence:

$$\mathcal{L}_w(\mathbf{x}) = \mathbb{E}_{p_w(t)}\left[\mathcal{L}(t; \mathbf{x})\right] + \text{constant} \tag{12}$$

where $p_w(t)$ is a probability distribution determined by the weighting function, namely $p_w(t) := (d/dt\, w(\lambda_t))$, with support on $t \in [0,1]$. The probability distribution $p_w(t)$ has Dirac delta peak of typically very small mass $w(\lambda_{\max})$ at $t = 0$.

Note that:

$$\mathcal{L}(t; \mathbf{x}) = D_{KL}(q(\mathbf{z}_{t,\ldots,1}|\mathbf{x})||p(\mathbf{z}_{t,\ldots,1})) \tag{13}$$
$$\geq D_{KL}(q(\mathbf{z}_t|\mathbf{x})||p(\mathbf{z}_t)) = -\mathbb{E}_{q(\mathbf{z}_t|\mathbf{x})}[\log p(\mathbf{z}_t)] + \text{constant}. \tag{14}$$

More specifically, $\mathcal{L}(t; \mathbf{x})$ equals the expected negative ELBO of noise-perturbed data, plus a constant; see Section C for a detailed derivation.

This concludes our proof of Theorem 1. ∎

This result provides us the new insight that any of the objectives with (implied) monotonic weighting, as listed in Table 1, can be understood as equivalent to the ELBO with simple data augmentation, namely additive noise. Specifically, this is a form of Distribution Augmentation (DistAug), where the model is conditioned on the augmentation indicator during training, and conditioned on 'no augmentation' during sampling.

Monotonicity of $w(\lambda)$ holds for a number of models with state-of-the-art perceptual quality, including VoiceBox for speech generation [Le et al., 2023], and Simple Diffusion for image generation [Hoogeboom et al., 2023].

### 4.1 Understanding the effect of weighting functions on perceptual quality

In this subsection, we propose possible explanations of the reason that the weighted objective, or equivalently the ELBO with data augmentation assuming monotonic weightings, can effectively improve perceptual quality.

**Connection to low-bit training.** Earlier work [Kingma and Dhariwal, 2018] found that training on 5-bit data, removing the three least significant bits from the original 8-bit data, can lead to higher fidelity models. A likely reason is the more significant bits are also more important to human perception, and removing the three least significant bits from the data allows the model to spend more capacity on modeling the more significant bits. In [Kingma and Dhariwal, 2018], training on 5-bit data was performed by adding uniform noise (with the appropriate scale) to the data before feeding it to the model, but it was found that adding Gaussian noise had a similar effect. Adding Gaussian noise with a single noise level is a special case of the weighted objective where the weighting function is a step function. The effect of uniform noise can be approximated with a sigmoidal weighting function; in Appendix I we dive into more details.

**Fourier analysis.** To better understand why additive Gaussian noise can improve perceptual quality, consider the Fourier transform (FT) of a clean natural image perturbed witjh additive Gaussian noise. Since the FT is a linear transformation, the FT of a the sum of a natural image and Gaussian noise, equals the sum of the FT of the clean image and the FT of the Gaussian noise. For natural images, it is known that the power spectrum, i.e. the average magnitude of the FT as a function of spatial frequency, decreases fast as the frequency increases [Burton and Moorhead, 1987], which means that the lowest frequencies have by far the highest magnitudes. On the other hand, the power spectrum of a Gaussian white noise is roughly constant. When adding noise to a clean natural image, the signal-to-noise ratio (i.e. the ratio of the power spectra of the clean and Gaussian noise images) in high frequency regions is lower. As a result, additive Gaussian noise effectively destroys high-frequency information in the data more quickly than the low-frequency information, pushing the model to focus more on the low frequency components of the data, which often correspond to high-level content and global structure that is more crucial to perception. See Appendix J for a more detailed discussion.

| Model parameterization | Training noise schedule | Weighting function | Monotonic? | DDPM sampler | | EDM sampler | |
|---|---|---|---|---|---|---|---|
| | | | | FID ↓ | IS ↑ | FID ↓ | IS ↑ |
| $\epsilon$-parametrization | Cosine | sech$(\lambda/2)$ (Baseline) | | 1.85 | 54.1 ± 0.79 | 1.55 | 59.2 ± 0.78 |
| " | Cosine | sigmoid$(-\lambda + 1)$ | ✓ | 1.75 | 55.3 ± 1.23 | | |
| " | Cosine | sigmoid$(-\lambda + 2)$ | ✓ | **1.68** | **56.8** ± 0.85 | 1.46 | 60.4 ± 0.86 |
| " | Cosine | sigmoid$(-\lambda + 3)$ | ✓ | 1.73 | 56.1 ± 1.36 | | |
| " | Cosine | sigmoid$(-\lambda + 4)$ | ✓ | 1.80 | 55.1 ± 1.65 | | |
| " | Cosine | sigmoid$(-\lambda + 5)$ | ✓ | 1.94 | 53.5 ± 1.12 | | |
| " | Adaptive | sigmoid$(-\lambda + 2)$ | ✓ | 1.70 | 54.8 ± 1.20 | **1.44** | 60.6 ± 1.44 |
| " | Adaptive | EDM-monotonic | ✓ | **1.67** | **56.8** ± 0.90 | **1.44** | **61.1** ± 1.80 |
| EDM [Karras et al., 2022] | EDM (training) | EDM (Baseline) | | | | 1.36 | |
| EDM (our reproduction) | EDM (training) | EDM (Baseline) | | | | 1.45 | 60.7 ± 1.19 |
| " | Adaptive | EDM | | | | **1.43** | 63.2 ± 1.76 |
| " | Adaptive | sigmoid$(-\lambda + 2)$ | ✓ | | | 1.55 | **63.7** ± 1.14 |
| " | Adaptive | EDM-monotonic | ✓ | | | **1.43** | **63.7** ± 1.48 |
| $\mathbf{v}$-parametrization | Adaptive | $\exp(-\lambda/2)$ (Baseline) | ✓ | | | 1.62 | 58.0 ± 1.56 |
| " | Adaptive | sigmoid$(-\lambda + 2)$ | ✓ | | | 1.51 | 64.4 ± 1.28 |
| " | Adaptive | EDM-monotonic | ✓ | | | **1.45** | **64.6** ± 1.35 |

Table 2: ImageNet 64x64 results. See Section 5.2.

# 5 Experiments

Inspired by the theoretical results in Section 4, in this section we propose several monotonic weighting functions, and present experiments that test the effectiveness of the monotonic weighting functions compared to baseline (non-monotonic) weighting functions. In addition, we test the adaptive noise schedule (Section 3.2). For brevity we had to pick a name for our models. Since we build on earlier results on Variational Diffusion Models (VDM) [Kingma et al., 2021], and our objective is equivalent to the VDM objective combined with data augmentation, we name our models *VDM++*.

## 5.1 Weighting functions

We experimented with several new monotonic weighting functions. First, a sigmoidal weighting of the form sigmoid$(-\lambda + k)$, inspired by the sigmoidal weighting corresponding to low-bit training (see Section 4.1 and Appendix I for details), where $k$ is a hyper-parameter. In addition, inspired by the EDM weighting function from Karras et al. [2022] (non-monotonic, Table 1), we test a weighting function indicated with 'EDM-monotonic', which is identical to the EDM weighting, except that it is made monotonic by letting $w(\lambda) = \max_\lambda \tilde{w}(\lambda)$ for $\lambda < \arg\max_\lambda \tilde{w}(\lambda)$, where $\tilde{w}(\lambda)$ indicates the original EDM weighting function. Put simply, 'EDM-monotonic' equals the EDM weighting, except that it is constant to the left of its peak; see Figure 3.

## 5.2 ImageNet 64x64

All experiments on ImageNet 64x64 were done with the U-Net diffusion model architecture from [Nichol and Dhariwal, 2021]. We carried out extensive ablation studies over several design choices of diffusion models, namely model parametrization, training noise schedules, weighting functions and samplers. For $\epsilon$-parametrization model, we took iDDPM [Nichol and Dhariwal, 2021] as the baseline, which utilized cosine noise schedule and a non-monotonic sech$(\lambda/2)$ weighting. For EDM parmetrization, we recruited the setting in Karras et al. [2022] as the baseline, with EDM training noise schedule (Figure 2) and non-monotonic EDM weighting (Figure 3). For $\mathbf{v}$-parametrization model, we followed Salimans and Ho [2022] taking the $\mathbf{v}$-prediction loss as the baseline, which leads to a monotonic $\exp(-\lambda/2)$ weighting. We always use adaptive noise schedule under this setting. We started by searching the optimal hyperparameter for the sigmoidal weighting under the $\epsilon$-parametrization model, and then applied the best sigmoidal weighting and EDM-monotonic weighting under other settings. We evaluated models with two samplers, DDPM and EDM samplers, with their corresponding sampling noise schedules. Table 2 summarizes the FID [Heusel et al., 2017] and Inception scores [Salimans et al., 2016] across different settings. From the table we made several observations as follows.

First, our adaptive noise schedule (Appendix F) works equally well as the hand-tuned fixed noise schedules. We verified this under $\epsilon$-parametrization with sigmoid($-\lambda + 2$) weighting and EDM parametrization with EDM weighting. Under both settings, the adaptive noise schedule results in slightly better FID and Inception scores with the EDM sampler, and slightly worse scores with the DDPM sampler for the former setting. Although the adaptive noise schedule did not significantly affect the end result, it allows us to experiment with new weighting functions while avoiding hand-tuning of the noise schedule. In addition, it sometimes leads to faster training; see Appendix F.

Second, it is possible to modify an existing non-monotonic weighting function to a monotonic one with the minimal change, and results in on-par sample quality. Specifically, under EDM parameterization and adaptive noise schedule, we change EDM weighting to EDM-monotonic weighting with the very straightforward approach stated in Section 5.1. It results in nearly the same FID score and slightly better Inception score.

Third, our proposed new weighting functions, sigmoid($-\lambda + 2$) and EDM monotonic weightings, work better than the baseline settings across different model parameterization and samplers in most cases. Larger performance gain has been observed under $\epsilon$-parameterization and $\mathbf{v}$-parameterization settings. Our-reimplementation of the EDM parameterization baseline setting could not exactly reproduce their reported FID number (1.36), but comes close (1.45). We observed less significant improvement under this setting, possibly because their already exhaustive search of the design space.

Finally, the above observations remain consistent across different model parameterization and samplers, indicating the generalizability of our proposed weighting functions and noise schedules.

## 5.3 High resolution ImageNet

In our final experiments, we tested whether the weighting functions that resulted in the best scores on ImageNet 64×64, namely sigmoid($-\lambda + 2$) and 'EDM-monotonic', also results in competitive scores on high-resolution generation. As baseline we use the *Simple Diffusion* model from Hoogeboom et al. [2023], which reported the best FID scores to date on high-resolution ImageNet without sampling modifications (e.g. guidance).

We recruited the large U-ViT model from Simple Diffusion [Hoogeboom et al., 2023], and changed the training noise schedule and weighting function to our proposed ones. Note that for higher-resolution models, Hoogeboom et al. [2023] proposed a shifted version of the cosine noise schedule (Table 4), that leads to a shifted version of the weighting function $w(\lambda)$. Similarly, we extended our proposed sigmoidal and 'EDM-monotonic' weightings to their shifted versions (see Appendix D.2.1 for details). For fair comparison, we adopted the same vanilla DDPM sampler as Simple Diffusion, without other advanced sampling techniques such as second-order sampling or rejection sampling. As shown in Table 3, with our adaptive noise schedule for training, the two weighting functions we proposed led to either better or comparable FID and IS scores on ImageNet 128×128, compared to the baseline Simple Diffusion approach.

Next, we test our approach on ImageNet generation of multiple high resolutions (i.e., resolutions 128, 256 and 512), and compare with existing methods in the literature. See Table 3 for the summary of quantitative evaluations and Figure 1 for some generated samples by our approach. With the shifted version of 'EDM-monotonic' weighting, we achieved state-of-the-art FID and IS scores on all three resolutions of ImageNet generation among all approaches without guidance. With classifier-free guidance (CFG) [Ho and Salimans, 2022], our method outperforms all diffusion-based approaches on resolutions 128 and 512. On resolution 256, our method only falls a bit behind Gao et al. [2023] and Hang et al. [2023], both of which were build upon the latent space of a pretrained auto-encoder from *latent diffusion models* [Rombach et al., 2022a] that was trained on much larger image datasets than ImageNet, while our model was trained on the ImageNet dataset only. It is worth noting that we achieve significant improvements compared to Simple Diffusion which serves as the backbone of our method, on all resolutions, with and without guidance. It is possible to apply our proposed weighting functions and adaptive noise schedules to other diffusion-based approaches such as Gao et al. [2023] to further improve their performance, which we shall leave to the future work.

| Method | Without guidance FID ↓ | | | With guidance FID ↓ | | |
|---|---|---|---|---|---|---|
| | train | eval | IS ↑ | train | eval | IS ↑ |
| **128 × 128 resolution** | | | | | | |
| ADM [Dhariwal and Nichol, 2022] | 5.91 | | | 2.97 | | |
| CDM [Ho et al., 2022] | 3.52 | 3.76 | 128.8 ± 2.5 | | | |
| RIN [Jabri et al., 2022] | 2.75 | | 144.1 | | | |
| Simple Diffusion (U-Net) [Hoogeboom et al., 2023] | 2.26 | 2.88 | 137.3 ± 2.0 | | | |
| Simple Diffusion (U-ViT, L) [Hoogeboom et al., 2023] | 1.91 | 3.23 | 171.9 ± 2.5 | 2.05 | 3.57 | 189.9 ± 3.5 |
| **VDM++ (Ours)**, $w(\lambda) = \text{sigmoid}(-\lambda + 2)$ | 1.91 | 3.41 | **183.1** ± 2.2 | | | |
| **VDM++ (Ours)**, EDM-monotonic weighting | **1.75** | **2.88** | 171.1 ± 2.7 | **1.78** | **3.16** | **190.5** ± 2.3 |
| **256 × 256 resolution** | | | | | | |
| BigGAN-deep (no truncation) [Brock et al., 2018] | 6.9 | | 171.4 ± 2.0 | | | |
| MaskGIT [Chang et al., 2022] | 6.18 | | 182.1 | | | |
| ADM [Dhariwal and Nichol, 2022] | 10.94 | | | 3.94 | | 215.9 |
| CDM [Ho et al., 2022] | 4.88 | 4.63 | 158.7 ± 2.3 | | | |
| RIN [Jabri et al., 2022] | 3.42 | | 182.0 | | | |
| Simple Diffusion (U-Net) [Hoogeboom et al., 2023] | 3.76 | 3.71 | 171.6 ± 3.1 | | | |
| Simple Diffusion (U-ViT, L) [Hoogeboom et al., 2023] | 2.77 | 3.75 | 211.8 ± 2.9 | 2.44 | 4.08 | 256.3 ± 5.0 |
| **VDM++ (Ours)**, EDM-monotonic weighting | **2.40** | **3.36** | **225.3** ± 3.2 | **2.12** | **3.69** | **267.7** ± 4.9 |
| *Latent diffusion with pretrained VAE:* | | | | | | |
| DiT-XL/2 [Peebles and Xie, 2022] | 9.62 | | 121.5 | 2.27 | | 278.2 |
| U-ViT [Bao et al., 2023] | | | | 3.40 | | |
| Min-SNR-$\gamma$ [Hang et al., 2023] | | | | 2.06 | | |
| MDT [Gao et al., 2023] | 6.23 | | 143.0 | 1.79 | | 283.0 |
| **512 × 512 resolution** | | | | | | |
| MaskGIT [Chang et al., 2022] | 7.32 | | 156.0 | | | |
| ADM [Dhariwal and Nichol, 2022] | 23.24 | | | 3.85 | | 221.7 |
| RIN [Jabri et al., 2022] | | | | 3.95 | | 216.0 |
| Simple Diffusion (U-Net) [Hoogeboom et al., 2023] | 4.30 | 4.28 | 171.0 ± 3.0 | | | |
| Simple Diffusion (U-ViT, L) [Hoogeboom et al., 2023] | 3.54 | 4.53 | 205.3 ± 2.7 | 3.02 | 4.60 | 248.7 ± 3.4 |
| **VDM++ (Ours)**, EDM-monotonic weighting | **2.99** | **4.09** | **232.2** ± 4.2 | **2.65** | **4.43** | **278.1** ± 5.5 |
| *Latent diffusion with pretrained VAE:* | | | | | | |
| DiT-XL/2 [Peebles and Xie, 2022] | 12.03 | | 105.3 | 3.04 | | 240.8 |
| LDM-4 [Rombach et al., 2022a] | 10.56 | | 103.5 ± 1.2 | 3.60 | | 247.7 ± 5.6 |

Table 3: Comparison to approaches in the literature for high-resolution ImageNet generation. 'With guidance' indicates that the method was combined with classifier-free guidance [Ho and Salimans, 2022]. [†] Models under 'Latent diffusion with pretrained VAE' use the pre-trained VAE from Stable Diffusion [Rombach et al., 2022a], which used a much larger training corpus than the other models in this table.

# 6   Conclusion and Discussion

In summary, we have shown that the weighted diffusion loss, which generalizes diffusion objectives in the literature, has an interpretation as a weighted integral of ELBO objectives, with one ELBO per noise level. If the weighting function is monotonic, then we show that the objective has an interpretation as the ELBO objective with data augmentation, where the augmentation is additive Gaussian noise, with a distribution of noise levels.

Our results open up exciting new directions for future work. The newfound equivalence between monotonically weighted diffusion objectives and the ELBO with data augmentation allows for a direct apples-to-apples comparison of diffusion models with other likelihood-based models. For example, it allows one to optimize other likelihood-based models, such as autoregressive models, towards the same objective as monotonically weighted diffusion models. This would shine light on whether diffusion models are better or worse than other model types, as measured in terms of their held-out objectives as opposed to FID scores. We leave such interesting experiments to future work.

## Acknowledgments

We'd like to thank Alex Alemi and Ben Poole for fruitful discussions and feedback on early drafts. We thank Emiel Hoogeboom for advice and help on the implementation of Simple Diffusion.

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

# A Main proof

Here we'll provide a proof of Equation 11.

Note that like in the main text, we use shorthand notation:

$$\mathcal{L}(t; \mathbf{x}) := D_{KL}(q(\mathbf{z}_{t,\dots,1}|\mathbf{x})||p(\mathbf{z}_{t,\dots,1})) \tag{15}$$

## A.1 Time derivative of $D_{KL}(q(\mathbf{z}_{t,\dots,1}|\mathbf{x})||p(\mathbf{z}_{t,\dots,1}))$

Let $dt$ denote an infinitesimal change in time. Note that $\mathcal{L}(t - dt; \mathbf{x})$ can be decomposed as the sum of a KL divergence and an expected KL divergence:

$$\mathcal{L}(t - dt; \mathbf{x}) = \mathcal{L}(t; \mathbf{x}) + \mathbb{E}_{q(\mathbf{z}_t|\mathbf{x})}[D_{KL}(q(\mathbf{z}_{t-dt}|\mathbf{z}_t, \mathbf{x})||p(\mathbf{z}_{t-dt}|\mathbf{z}_t)] \tag{16}$$

Due to this identity, the time derivative $d/dt\, \mathcal{L}(t; \mathbf{x})$ can be expressed as:

$$\frac{d}{dt}\mathcal{L}(t; \mathbf{x}) = \frac{1}{dt}(\mathcal{L}(t; \mathbf{x}) - \mathcal{L}(t - dt; \mathbf{x})) \tag{17}$$

$$= -\frac{1}{dt}\mathbb{E}_{q(\mathbf{z}_t|\mathbf{x})}[D_{KL}(q(\mathbf{z}_{t-dt}|\mathbf{z}_t, \mathbf{x})||p(\mathbf{z}_{t-dt}|\mathbf{z}_t)] \tag{18}$$

In Appendix E of [Kingma et al., 2021], it is shown that in our model, this equals:

$$\frac{d}{dt}\mathcal{L}(t; \mathbf{x}) = -\frac{1}{2}\frac{SNR(t - dt) - SNR(t)}{dt}||\mathbf{x} - \hat{\mathbf{x}}_{\boldsymbol{\theta}}(\mathbf{z}_t; \lambda_t)||_2^2 \tag{19}$$

$$= \frac{1}{2}SNR'(t)||\mathbf{x} - \hat{\mathbf{x}}_{\boldsymbol{\theta}}(\mathbf{z}_t; \lambda_t)||_2^2 \tag{20}$$

where $\mathbf{z}_t = \alpha_\lambda \mathbf{x} + \sigma_\lambda \boldsymbol{\epsilon}$, and $SNR(t) := \exp(\lambda)$ in our model, and $SNR'(t) = d/dt\, SNR(t) = e^\lambda\, d\lambda/dt$, so in terms of our definition of $\lambda$, this is:

$$\frac{d}{dt}\mathcal{L}(t; \mathbf{x}) = \frac{1}{2}e^\lambda \frac{d\lambda}{dt}\mathbb{E}_{\boldsymbol{\epsilon} \sim \mathcal{N}(0, \mathbf{I})}\left[||\mathbf{x} - \hat{\mathbf{x}}_{\boldsymbol{\theta}}(\mathbf{z}_t; \lambda_t)||_2^2\right] \tag{21}$$

In terms of $\boldsymbol{\epsilon}$-prediction (see Section E.2), because $||\boldsymbol{\epsilon} - \hat{\boldsymbol{\epsilon}}_{\boldsymbol{\theta}}||_2^2 = e^\lambda||\mathbf{x} - \hat{\mathbf{x}}_{\boldsymbol{\theta}}||_2^2$ this simplifies to:

$$\frac{d}{dt}\mathcal{L}(t; \mathbf{x}) = \frac{1}{2}\frac{d\lambda}{dt}\mathbb{E}_{\boldsymbol{\epsilon} \sim \mathcal{N}(0, \mathbf{I})}\left[||\boldsymbol{\epsilon} - \hat{\boldsymbol{\epsilon}}_{\boldsymbol{\theta}}(\mathbf{z}_t; \lambda_t)||_2^2\right] \tag{22}$$

where $\mathbf{z}_\lambda = \alpha_\lambda \mathbf{x} + \sigma_\lambda \boldsymbol{\epsilon}$. This can be easily translated to other parameterizations; see E.2.

This allows us to rewrite the weighted loss of Equation 5 as:

$$\mathcal{L}_w(\mathbf{x}) = \frac{1}{2}\mathbb{E}_{t \sim \mathcal{U}(0,1), \boldsymbol{\epsilon} \sim \mathcal{N}(0, \mathbf{I})}\left[w(\lambda_t) \cdot -\frac{d\lambda}{dt} \cdot ||\hat{\boldsymbol{\epsilon}}_{\boldsymbol{\theta}}(\mathbf{z}_t; \lambda) - \boldsymbol{\epsilon}||_2^2\right] \tag{23}$$

$$= \mathbb{E}_{t \sim \mathcal{U}(0,1)}\left[w(\lambda_t) \cdot -\frac{1}{2}\frac{d\lambda}{dt}\mathbb{E}_{\boldsymbol{\epsilon} \sim \mathcal{N}(0, \mathbf{I})}\left[||\hat{\boldsymbol{\epsilon}}_{\boldsymbol{\theta}}(\mathbf{z}_t; \lambda) - \boldsymbol{\epsilon}||_2^2\right]\right] \tag{24}$$

$$= \mathbb{E}_{t \sim \mathcal{U}(0,1)}\left[-\frac{d}{dt}\mathcal{L}(t; \mathbf{x})\, w(\lambda_t)\right] \tag{25}$$

$$= \int_0^1 -\frac{d}{dt}\mathcal{L}(t; \mathbf{x})\, w(\lambda_t)\, dt \tag{26}$$

## A.2 Integration by parts

Integration by parts is a basic identity, which tells us that:

$$-\int_a^b f(t)g'(t)dt = \int_a^b f'(t)g(t)dt + f(a)g(a) - f(b)g(b)$$

This allows us to further rewrite the expression of the weighted loss in Equation 26 as:

$$\mathcal{L}_w(\mathbf{x}) = \int_0^1 -\frac{d}{dt}\mathcal{L}(t;\mathbf{x})\,w(\lambda_t)\,dt \tag{27}$$

$$= \int_0^1 \frac{d}{dt}w(\lambda_t)\,\mathcal{L}(t;\mathbf{x})\,dt + w(\lambda_{\max})\mathcal{L}(0;\mathbf{x}) - w(\lambda_{\min})\mathcal{L}(1;\mathbf{x}) \tag{28}$$

where

$$-w(\lambda_{\min})\,\mathcal{L}(1;\mathbf{x}) = -w(\lambda_{\min})\,D_{KL}(q(\mathbf{z}_1|\mathbf{x})||p(\mathbf{z}_1)) \tag{29}$$

is constant w.r.t. $\boldsymbol{\theta}$, since it does not involve the score function, and typically very small, since $D_{KL}(q(\mathbf{z}_1|\mathbf{x})||p(\mathbf{z}_1))$ is typically small by design.

The term $w(\lambda_{\max})D_{KL}(q(\mathbf{z}_{0,\ldots,1}|\mathbf{x})||p(\mathbf{z}_{0,\ldots,1}))$ is typically small, since $w(\lambda_{\max})$ is typically very small (see Figure 3).

This concludes our proof of Equation 11. ∎

## B   Visualization

We tried to create a helpful visualization of the result from Section A.2. Note that we can rewrite:

$$\int_0^1 \frac{d}{dt}w(\lambda_t)\,\mathcal{L}(t;\mathbf{x})\,dt = \int_{t=1}^{t=0} w(\lambda_t)\,d\mathcal{L}(t;\mathbf{x}) \tag{30}$$

The relationship in Equation 28 can be rewritten as:

$$w(\lambda_{\min})\,\mathcal{L}(1) + \int_{t=1}^{t=0} w(\lambda_t)\,d\mathcal{L}(t;\mathbf{x}) = w(\lambda_{\max})\,\mathcal{L}(0) + \int_{t=0}^{t=1} \mathcal{L}(t;\mathbf{x})\,dw(\lambda_t) \tag{31}$$

The first LHS term equals a weighted prior loss term, and the second LHS term equals the weighted diffusion loss. From a geometric perspective, the two LHS terms together define an area that equals the area given by the right term, as illustrated in the figure below.

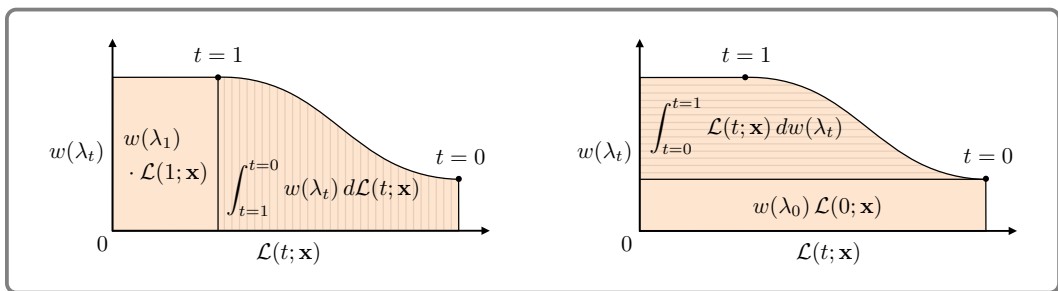

On the left, we have a rectangular area that equals a weighted prior loss $w(\lambda_{\min})\mathcal{L}(1)$, plus a curved area equal to the weighted diffusion loss $\int_{t=1}^{t=0} w(\lambda_t)\,d\mathcal{L}(t;\mathbf{x})$. This integral can be intuitively understood as a Riemann sum over many tiny rectangles going from left ($t=1$) to right ($t=0$), each with height $w(\lambda_t)$ and width $d\mathcal{L}(t;\mathbf{x})$. On the right, we have the same total area, but divided up into two different subareas: a rectangular area $w(\lambda_{\max})\mathcal{L}(0)$ and a curved area that equals the integral $\int_{t=0}^{t=1} \mathcal{L}(t;\mathbf{x})\,dw(\lambda_t)$ going upwards from $t=0$ to $t=1$, which can also be intuitively understood as another Riemann sum, with each tiny rectangle having width $\mathcal{L}(t;\mathbf{x})$ and height $dw(\lambda_t)$. The area of each of those tiny rectangles on the right can be understood as the ELBO at each noise level, $\mathcal{L}(t;\mathbf{x})$, times the weight of the ELBO at each noise level, $dw(\lambda_t)$.

## C   Relationship between $D_{KL}(q(\mathbf{z}_{t,\ldots,1}|\mathbf{x})||p(\mathbf{z}_{t,\ldots,1}))$ and the ELBO

First, note that:

$$\mathcal{L}(t;\mathbf{x}) = D_{KL}(q(\mathbf{z}_{t,\ldots,1}|\mathbf{x})||p(\mathbf{z}_{t,\ldots,1})) \geq D_{KL}(q(\mathbf{z}_t|\mathbf{x})||p(\mathbf{z}_t)) \tag{32}$$

More precisely, the joint KL divergence $D_{KL}(q(\mathbf{z}_{t,\dots,1}|\mathbf{x})||p(\mathbf{z}_{t,\dots,1}))$ is the expected negative ELBO of noise-perturbed data, plus a constant entropy term:

$$\mathcal{L}(t;\mathbf{x}) = D_{KL}(q(\mathbf{z}_{t,\dots,1}|\mathbf{x})||p(\mathbf{z}_{t,\dots,1})) = -\mathbb{E}_{q(\mathbf{z}_t|\mathbf{x})}[\text{ELBO}_t(\mathbf{z}_t)] - \underbrace{\mathcal{H}(q(\mathbf{z}_t|\mathbf{x}))}_{\text{constant}} \quad (33)$$

where the ELBO of noise-perturbed data is:

$$\text{ELBO}_t(\mathbf{z}_t) := \mathbb{E}_{q(\tilde{\mathbf{z}}_t|\mathbf{z}_t)}[\log p(\mathbf{z}_t, \tilde{\mathbf{z}}_t) - \log q(\tilde{\mathbf{z}}_t|\mathbf{z}_t)] \quad (34)$$

$$\leq \log p(\mathbf{z}_t) \quad (35)$$

where $\tilde{\mathbf{z}}_t := \mathbf{z}_{t+dt,\dots,1}$.

So, $\mathcal{L}(t;\mathbf{x})$ is the expected negative ELBO of noise-perturbed data $\mathbf{z}_t$:

$$\mathcal{L}(t;\mathbf{x}) = -\mathbb{E}_{q(\mathbf{z}_t|\mathbf{x})}[\text{ELBO}_t(\mathbf{z}_t)] + \text{constant} \geq -\mathbb{E}_{q(\mathbf{z}_t|\mathbf{x})}[\log p(\mathbf{z}_t)] + \text{constant} \quad (36)$$

Therefore, the expression of the weighted loss in Equation 12 can be rewritten as:

$$\mathcal{L}_w(\mathbf{x}) = \mathbb{E}_{p_w(t)}[\mathcal{L}(t;\mathbf{x})] + \text{constant} \quad (37)$$

$$= -\underbrace{\mathbb{E}_{p_w(t),q(\mathbf{z}_t|\mathbf{x})}[\text{ELBO}_t(\mathbf{z}_t)]}_{\text{ELBO of noise-perturbed data}} + \text{constant} \quad (38)$$

$$\geq -\underbrace{\mathbb{E}_{p_w(t),q(\mathbf{z}_t|\mathbf{x})}[\log p(\mathbf{z}_t)]}_{\text{Log-likelihood of noise-perturbed data}} + \text{constant} \quad (39)$$

where $w(\lambda_{\min})$ is constant w.r.t. the diffusion model parameters. Therefore, minimizing $\mathcal{L}_w(\mathbf{x})$ is equivalent to maximizing this expected ELBO of noise-perturbed data.

# D    Derivation of weighting functions for previous works

The loss function used in previous works are equivalent to the weighted loss with a certain choice of noise schedule and weighting function $w(\lambda)$. In this section, we derive these weighting functions $w(\lambda)$.

### D.1    'Elucidating Diffusion Models' (EDM) [Karras et al., 2022]

Karras et al. [2022] proposes the following training objective:

$$\mathcal{L}_{\text{edm}}(\mathbf{x}) = \mathbb{E}_{\tilde{\sigma}\sim p(\tilde{\sigma}),\boldsymbol{\epsilon}\sim\mathcal{N}(0,\mathbf{I})}\left[\tilde{w}(\tilde{\sigma})\|\mathbf{x} - \hat{\mathbf{x}}_{\boldsymbol{\theta}}(\mathbf{z}_{\tilde{\sigma}};\tilde{\sigma})\|_2^2\right], \quad (40)$$

where $p(\tilde{\sigma})$ and $\tilde{w}(\tilde{\sigma})$ are defined as:

$$p(\log\tilde{\sigma}) = \mathcal{N}(\log\tilde{\sigma}; P_{\text{mean}}, P_{\text{std}}^2), \quad (41)$$

$$\tilde{w}(\tilde{\sigma}) = (\tilde{\sigma}^2 + \tilde{\sigma}_{\text{data}}^2)/(\tilde{\sigma}^2 \cdot \tilde{\sigma}_{\text{data}}^2), \quad (42)$$

where $\tilde{\sigma}$ is equivalent to the standard deviation of noise added to the clean sample $\mathbf{x}$ in the VP SDE case, so $\tilde{\sigma}^2 = e^{-\lambda}$. Karras et al. [2022] used hyperparameters $P_{\text{mean}} = -1.2$, $P_{\text{std}} = 1.2$ and $\tilde{\sigma}_{\text{data}} = 0.5$.

This can be rewritten in terms of $\lambda = -2\log\tilde{\sigma}$ as:

$$\mathcal{L}_{\text{edm}}(\mathbf{x}) = \mathbb{E}_{p(\lambda),\boldsymbol{\epsilon}\sim\mathcal{N}(0,\mathbf{I})}\left[\tilde{w}(\lambda)\|\mathbf{x} - \hat{\mathbf{x}}_{\boldsymbol{\theta}}(\mathbf{z}_\lambda;\lambda)\|_2^2\right], \quad (43)$$

where $p(\lambda)$ and $\tilde{w}(\lambda)$ are defined as

$$p(\lambda) = \mathcal{N}(\lambda; 2.4, 2.4^2), \quad (44)$$

$$\tilde{w}(\lambda) = (e^{-\lambda} + 0.5^2)/(e^{-\lambda} \cdot 0.5^2), \quad (45)$$

Comparing Equation 45 with the weighted loss expressed in terms of $\mathbf{x}$-prediction parameterization (Section E.2), we see that (ignoring the constant scaling factor $1/2$) the EDM objective is a special case of the weighted loss with weighting function:

$$w(\lambda) = p(\lambda)e^{-\lambda} \cdot \tilde{w}(\lambda) \quad (46)$$

$$= p(\lambda)(e^{-\lambda} + 0.5^2)/0.5^2. \quad (47)$$

where the divison by $\tilde{\sigma}_{\text{data}}^2$ can be ignored since it's constant. This leads to:

$$w(\lambda) = \mathcal{N}(\lambda; 2.4, 2.4^2)(e^{-\lambda} + 0.5^2) \quad (48)$$

## D.2 The v-prediction loss / 'SNR+1'-weighting [Salimans and Ho, 2022]

Salimans and Ho [2022] introduced the **v**-parameterization, with a **v**-prediction model $\hat{\mathbf{v}}$, where:

$$\mathbf{v} := \alpha_\lambda \boldsymbol{\epsilon} - \sigma_\lambda \mathbf{x} \tag{49}$$

$$\hat{\mathbf{v}} := \alpha_\lambda \hat{\boldsymbol{\epsilon}} - \sigma_\lambda \hat{\mathbf{x}} \tag{50}$$

They propose to minimize a **v**-prediction loss, $\mathbb{E}[||\mathbf{v} - \hat{\mathbf{v}}||_2^2]$. Note that $\hat{\boldsymbol{\epsilon}} = (\mathbf{z}_\lambda - \alpha_\lambda \hat{\mathbf{x}})/\sigma_\lambda$. For our general family, this implies:

$$||\mathbf{v} - \hat{\mathbf{v}}||_2^2 = \sigma_\lambda^2 (e^\lambda + 1)^2 ||\mathbf{x} - \hat{\mathbf{x}}||_2^2 \tag{51}$$

$$= \alpha_\lambda^2 (e^{-\lambda} + 1)^2 ||\boldsymbol{\epsilon} - \hat{\boldsymbol{\epsilon}}||_2^2 \tag{52}$$

In the special case of the variance preserving (VP) SDE, this simplifies to:

$$||\mathbf{v} - \hat{\mathbf{v}}||_2^2 = (e^\lambda + 1)||\mathbf{x} - \hat{\mathbf{x}}||_2^2 \tag{53}$$

$$= (e^{-\lambda} + 1)||\boldsymbol{\epsilon} - \hat{\boldsymbol{\epsilon}}||_2^2. \tag{54}$$

Since the $\boldsymbol{\epsilon}$-prediction loss corresponds to minimizing the weighted loss with $w(\lambda) = p(\lambda)$, the **v**-prediction loss corresponds to minimizing the weighted loss with $w(\lambda) = (e^{-\lambda} + 1)p(\lambda)$.

Note that Salimans and Ho [2022] view the loss from the **x**-prediction viewpoint, instead of our ELBO viewpoint; so in their view, minimizing simply $||\mathbf{x} - \hat{\mathbf{x}}||_2$ means no weighting. Note that $e^\lambda$ is the signal-to-noise ratio (SNR). Since $||\boldsymbol{\epsilon} - \hat{\boldsymbol{\epsilon}}||_2^2 = e^\lambda ||\mathbf{x} - \hat{\mathbf{x}}||_2^2$, they call the $\boldsymbol{\epsilon}$-prediction loss 'SNR weighting', and since $||\mathbf{v} - \hat{\mathbf{v}}||_2^2 = (e^\lambda + 1)||\mathbf{x} - \hat{\mathbf{x}}||_2^2$, they call this 'SNR+1'-weighting.

Salimans and Ho [2022] propose to use optimize a VP SDE with a cosine schedule $p(\lambda) = \text{sech}(\lambda/2)/(2\pi) = 1/(2\pi \cosh(-\lambda/2))$ and the **v**-prediction loss: $\mathbb{E}[||\mathbf{v} - \hat{\mathbf{v}}||_2^2]$. This corresponds to minimizing the weighted loss with:

$$w(\lambda) = (e^{-\lambda} + 1)p(\lambda) \tag{55}$$

$$= (e^{-\lambda} + 1)/(2\pi \cosh(-\lambda/2)) \tag{56}$$

$$= \pi e^{-\lambda/2} \tag{57}$$

The factor $\pi$ can be ignored since it's constant, so we can equivalently use:

$$w(\lambda) = e^{-\lambda/2} \tag{58}$$

### D.2.1 With shifted cosine schedule

[Hoogeboom et al., 2023] extended the cosine schedule to a shifted version: $p(\lambda) = \text{sech}(\lambda/2 - s)/(2\pi)$, where $s = \log(64/d)$, where 64 is the base resolution and $d$ is the model resolution (e.g. 128, 256, 512, etc.). In this case the weighting is:

$$w(\lambda) = (e^{-\lambda} + 1)p(\lambda) \tag{59}$$

$$= (2/\pi)e^{-s}e^{-\lambda/2} \tag{60}$$

Since $(2/\pi)e^{-s}$ is constant w.r.t $\lambda$, the weighting is equivalent to the weighting for the unshifted cosine schedule.

## D.3 Flow Matching with the Optimal Transport flow path (FM-OT) [Lipman et al., 2022]

Flow Matching [Lipman et al., 2022] with the Optimal Transport flow path can be seen as a special case of Gaussian diffusion with the weighted loss.

### D.3.1 Noise schedule

Note that in [Lipman et al., 2022], time goes from 1 to 0 as we go forward in time. Here, we'll let time go from 0 to 1 as we go forward in time, consistent with the rest of this paper. We'll also assume

$\sigma_0 = 0$, for which we can later correct by truncation (see Section E.3.1). In this model, the forward process $q(\mathbf{z}_t|\mathbf{x})$ is defined by:

$$\mathbf{z}_t = \alpha_t \mathbf{x} + \sigma_t \boldsymbol{\epsilon} \tag{61}$$

$$= (1-t)\mathbf{x} + t\boldsymbol{\epsilon} \tag{62}$$

This implies that the log-SNR is given by:

$$\lambda_t = f_\lambda(t) = \log(\alpha_t^2/\sigma_t^2) \tag{63}$$

$$= 2\log((1-t)/t) \tag{64}$$

Its inverse is given by:

$$t = f_\lambda^{-1}(\lambda) = 1/(1 + e^{\lambda/2}) \tag{65}$$

$$= \text{sigmoid}(-\lambda/2) \tag{66}$$

The derivative, as a function of $t$, is:

$$\frac{d\lambda}{dt} = \frac{d}{dt}f_\lambda(t) = 2/(-t + t^2) \tag{67}$$

This derivative of its inverse, as a function of $\lambda$, is:

$$\frac{dt}{d\lambda} = \frac{d}{d\lambda}f_\lambda^{-1}(\lambda) = \frac{d}{d\lambda}\text{sigmoid}(-\lambda/2) = -\text{sech}^2(\lambda/4)/8 \tag{68}$$

The corresponding density is

$$p(\lambda) = -\frac{d}{d\lambda}f_\lambda^{-1}(\lambda) = \text{sech}^2(\lambda/4)/8 \tag{69}$$

which is a Logistic distribution; see also Table 4.

### D.3.2 Score function parameterization and loss function

Lipman et al. [2022] then propose the following generative model ODE:

$$d\mathbf{z} = -\hat{\mathbf{o}}(\mathbf{z}_t, t)dt \tag{70}$$

The model is then optimized with the *Conditional flow matching* (CFM) loss:

$$\mathcal{L}_{\text{CFM}}(\mathbf{x}) = \mathbb{E}_{t\sim\mathcal{U}(0,1),\boldsymbol{\epsilon}\sim\mathcal{N}(0,\mathbf{I})}[||\mathbf{o} - \hat{\mathbf{o}}||_2^2] \tag{71}$$

where they use the parameterization:

$$\mathbf{o} := \mathbf{x} - \boldsymbol{\epsilon} \tag{72}$$

### D.3.3 Weighting function

What is the weighting function $w(\lambda)$ corresponding to this loss? Note that this parameterization means that:

$$\mathbf{z}_t = (1-t)\mathbf{x} + t\boldsymbol{\epsilon} \tag{73}$$

$$= (1-t)\mathbf{o} + \boldsymbol{\epsilon} \tag{74}$$

$$\mathbf{o} = (\mathbf{z}_t - \boldsymbol{\epsilon})/(1-t) \tag{75}$$

Since $t = 1/(1 + e^{\lambda/2})$, we have that $1/(1-t) = 1 + e^{-\lambda/2}$, so parameterized as a function of $\lambda$, we have:

$$\mathbf{o} = (\mathbf{z}_\lambda - \boldsymbol{\epsilon})(1 + e^{-\lambda/2}) \tag{76}$$

Likewise, we can parameterize $\mathbf{o}$-prediction in terms of $\boldsymbol{\epsilon}$-prediction:

$$\hat{\mathbf{o}}(\mathbf{z}_\lambda, \lambda) = (\mathbf{z}_\lambda - \hat{\boldsymbol{\epsilon}}(\mathbf{z}_\lambda, \lambda))(1 + e^{-\lambda/2}) \tag{77}$$

We can translate the $\mathbf{o}$-prediction loss to a $\boldsymbol{\epsilon}$-prediction loss:

$$||\mathbf{o} - \hat{\mathbf{o}}(\mathbf{z}_\lambda, \lambda)||_2^2 = (1 + e^{-\lambda/2})^2||\boldsymbol{\epsilon} - \hat{\boldsymbol{\epsilon}}(\mathbf{z}_\lambda, \lambda)||_2^2 \tag{78}$$

Therefore, combining the derivations above, the CFM loss, formulated in terms of the $\lambda$ parameterization instead of $t$, and in terms of the $\boldsymbol{\epsilon}$-prediction parameterization instead of the $\mathbf{o}$-prediction parameterization, is:

$$\mathcal{L}_{\text{CFM}}(\mathbf{x}) = \mathbb{E}_{t\sim\mathcal{U}(0,1),\boldsymbol{\epsilon}\sim\mathcal{N}(0,\mathbf{I})}[||\mathbf{o} - \hat{\mathbf{o}}(\mathbf{z}_t, t)||_2^2] \tag{79}$$

$$= \int_0^1 \mathbb{E}_{\boldsymbol{\epsilon}\sim\mathcal{N}(0,\mathbf{I})}[||\mathbf{o} - \hat{\mathbf{o}}(\mathbf{z}_t, t)||_2^2]\, dt \tag{80}$$

$$= \int_{\lambda_{\min}}^{\lambda_{\max}} -\frac{dt}{d\lambda}\mathbb{E}_{\boldsymbol{\epsilon}\sim\mathcal{N}(0,\mathbf{I})}[||\mathbf{o} - \hat{\mathbf{o}}(\mathbf{z}_\lambda, \lambda)||_2^2]\, d\lambda \tag{81}$$

$$= \int_{\lambda_{\min}}^{\lambda_{\max}} (\text{sech}^2(\lambda/4)/8)\mathbb{E}_{\boldsymbol{\epsilon}\sim\mathcal{N}(0,\mathbf{I})}[||\mathbf{o} - \hat{\mathbf{o}}(\mathbf{z}_\lambda, \lambda)||_2^2]\, d\lambda \tag{82}$$

$$= \int_{\lambda_{\min}}^{\lambda_{\max}} (\text{sech}^2(\lambda/4)/8)(1 + e^{-\lambda/2})^2\mathbb{E}_{\boldsymbol{\epsilon}\sim\mathcal{N}(0,\mathbf{I})}[||\boldsymbol{\epsilon} - \hat{\boldsymbol{\epsilon}}(\mathbf{z}_\lambda, \lambda)||_2^2]\, d\lambda \tag{83}$$

$$= \frac{1}{2}\int_{\lambda_{\min}}^{\lambda_{\max}} w(\lambda)\mathbb{E}_{\boldsymbol{\epsilon}\sim\mathcal{N}(0,\mathbf{I})}[||\boldsymbol{\epsilon} - \hat{\boldsymbol{\epsilon}}(\mathbf{z}_\lambda, \lambda)||_2^2]\, d\lambda \tag{84}$$

$$= \frac{1}{2}\mathbb{E}_{\boldsymbol{\epsilon}\sim\mathcal{N}(0,\mathbf{I}),\lambda\sim\tilde{p}(\lambda)}\left[\frac{w(\lambda)}{\tilde{p}(\lambda)}||\boldsymbol{\epsilon} - \hat{\boldsymbol{\epsilon}}(\mathbf{z}_\lambda, \lambda)||_2^2\right] \tag{85}$$

where $\tilde{p}$ is any distribution with full support on $[\lambda_{\min}, \lambda_{\max}]$, and where:

$$w(\lambda) = 2(\text{sech}^2(\lambda/4)/8)(1 + e^{-\lambda/2})^2 \tag{86}$$

$$= e^{-\lambda/2} \tag{87}$$

Therefore, this weighting is equivalent to the weighting for the $\mathbf{v}$-prediction loss with cosine schedule (Section D.2): the CFM loss is equivalent to the $\mathbf{v}$-prediction loss with cosine schedule.

### D.4 Inversion by Direct Iteration (InDI) [Delbracio and Milanfar, 2023]

Delbracio and Milanfar [2023] propose Inversion by Direct Iteration (InDI). Their forward process is identical to the forward process of FM-OT [Lipman et al., 2022] introduced in Section D.3:

$$\mathbf{z}_t = (1 - t)\mathbf{x} + t\boldsymbol{\epsilon} \tag{88}$$

As derived in Section D.3 above, this means that the distribution over log-SNR $\lambda$ is the Logistic distribution: $p(\lambda) = \text{sech}^2(\lambda/4)/8$. The proposed loss function is the $\mathbf{x}$-prediction loss:

$$\mathcal{L}_{\text{InDI}}(\mathbf{x}) = \mathbb{E}_{t\sim\mathcal{U}(0,1),\boldsymbol{\epsilon}\sim\mathcal{N}(0,\mathbf{I})}[||\mathbf{x} - \hat{\mathbf{x}}(\mathbf{z}_t, t)||_2^2] \tag{89}$$

Since $||\mathbf{x} - \hat{\mathbf{x}}||_2^2 = e^{-\lambda}||\boldsymbol{\epsilon} - \hat{\boldsymbol{\epsilon}}||_2^2$, and the $\boldsymbol{\epsilon}$-prediction loss corresponds to minimizing the weighted loss with $w(\lambda) = p(\lambda)$, the $\mathbf{x}$-prediction loss above corresponds to minimizing the weighted loss with:

$$w(\lambda) = e^{-\lambda}p(\lambda) \tag{90}$$

$$= e^{-\lambda}\text{sech}^2(\lambda/4)/8 \tag{91}$$

Which is a slightly different weighting then the FM-OT weighting, giving a bit more weighting to lower noise levels.

### D.5 Perception prioritized weighting (P2 weighting) [Choi et al., 2022]

Choi et al. [2022] proposed a new weighting function:

$$w(\lambda) = \frac{-dt/d\lambda}{(k + e^\lambda)^\gamma} = \frac{p(\lambda)}{(k + e^\lambda)^\gamma}, \tag{92}$$

where empirically they set $k = 1$ and $\gamma$ as either 0.5 or 1. Compared to the $\boldsymbol{\epsilon}$-prediction objective, where $w(\lambda) = dt/d\lambda = p(\lambda)$, this objective put more emphasis on the middle regime of the whole noise schedule, which Choi et al. [2022] hypothesized to be the most important regime for creating content that is sensitive to visual perception. When combined with the most commonly used cosine noise schedule [Nichol and Dhariwal, 2021], the weighting function becomes $w(\lambda) = \text{sech}(\lambda/2)/(1 + e^\lambda)^\gamma$.

### D.6 Min-SNR-$\gamma$ weighting [Hang et al., 2023]

Hang et al. [2023] proposed the following training objective:

$$\mathcal{L}_{\text{MinSNR}}(\mathbf{x}) = \mathbb{E}_{t \sim \mathcal{U}(0,1), \boldsymbol{\epsilon} \sim \mathcal{N}(0,\mathbf{I})} \left[ \min\{e^\lambda, \gamma\} \|\mathbf{x} - \hat{\mathbf{x}}(\mathbf{z}_t; \lambda)\|_2^2 \right] \tag{93}$$

$$= E_{t \sim \mathcal{U}(0,1), \boldsymbol{\epsilon} \sim \mathcal{N}(0,\mathbf{I})} \left[ \min\{1, \gamma e^{-\lambda}\} \|\boldsymbol{\epsilon} - \hat{\boldsymbol{\epsilon}}(\mathbf{z}_t; \lambda)\|_2^2 \right] \tag{94}$$

$$= E_{t \sim \mathcal{U}(0,1), \boldsymbol{\epsilon} \sim \mathcal{N}(0,\mathbf{I})} \left[ \min\{1, \gamma e^{-\lambda}\} \cdot -\frac{dt}{d\lambda} \cdot -\frac{d\lambda}{dt} \|\boldsymbol{\epsilon} - \hat{\boldsymbol{\epsilon}}(\mathbf{z}_t; \lambda)\|_2^2 \right] \tag{95}$$

$$= E_{t \sim \mathcal{U}(0,1), \boldsymbol{\epsilon} \sim \mathcal{N}(0,\mathbf{I})} \left[ \min\{1, \gamma e^{-\lambda}\} p(\lambda) \cdot -\frac{d\lambda}{dt} \|\boldsymbol{\epsilon} - \hat{\boldsymbol{\epsilon}}(\mathbf{z}_t; \lambda)\|_2^2 \right]. \tag{96}$$

Therefore, it corresponds to $w(\lambda) = \min\{1, \gamma e^{-\lambda}\} p(\lambda)$. The motivation of the work is to avoid the model focusing too much on small noise levels, since it shares similar hypothesis to [Choi et al., 2022] that small noise levels are responsible for cleaning up details that may not be perceptible. A cosine noise schedule is then combined with the proposed weighting function, leading to $w(\lambda) = \text{sech}(\lambda/2) \cdot \min\{1, \gamma e^{-\lambda}\}$. $\gamma$ is set as 5 empirically.

## E   Useful Equations

### E.1   SDEs

The forward process is a Gaussian diffusion process, whose time evolution is described by a stochastic differential equation (SDE):

$$d\mathbf{z} = \underbrace{\mathbf{f}(\mathbf{z}, t)}_{\text{drift}} dt + \underbrace{g(t)}_{\text{diffusion}} d\mathbf{w} \tag{97}$$

For derivations of diffusion SDEs, see Appendix B of [Song et al., 2021b]. Their $\beta(t)$ equals $\frac{d}{dt} \log(1 + e^{-\lambda_t})$ in our formulation, and their $\int_0^t \beta(s)ds$ equals $\log(1 + e^{-\lambda_t})$, where they assume that $\lambda \to \infty$ at $t = 0$.

### E.1.1   Variance-preserving (VP) SDE

A common choice is the variance-preserving (VP) SDE, which generalizes denoising diffusion models [Ho et al., 2020] to continuous time [Song et al., 2021b, Kingma et al., 2021]. In the VP case:

$$\mathbf{f}(\mathbf{z}, t) = -\frac{1}{2} \left( \frac{d}{dt} \log(1 + e^{-\lambda_t}) \right) \mathbf{z} \tag{98}$$

$$g(t)^2 = \frac{d}{dt} \log(1 + e^{-\lambda_t}) \tag{99}$$

$$\alpha_\lambda^2 = \text{sigmoid}(\lambda) \tag{100}$$

$$\sigma_\lambda^2 = \text{sigmoid}(-\lambda) \tag{101}$$

$$p(\mathbf{z}_1) = \mathcal{N}(0, \mathbf{I}) \tag{102}$$

### E.1.2   Variance-exploding (VE) SDE

Another common choice of the variance-exploding (VE) SDE. In the VE case:

$$\mathbf{f}(\mathbf{z}, t) = 0 \tag{103}$$

$$g(t)^2 = \frac{d}{dt} \log(1 + e^{-\lambda_t}) \tag{104}$$

$$\alpha_\lambda^2 = 1 \tag{105}$$

$$\sigma_\lambda^2 = e^{-\lambda} \tag{106}$$

$$p(\mathbf{z}_1) = \mathcal{N}(0, e^{-\lambda_{\min}} \mathbf{I}) \tag{107}$$

## E.2 Possible parameterizations of the score network

There are various ways of parameterizing the score network:

$$\mathbf{s}_{\boldsymbol{\theta}}(\mathbf{z}; \lambda) = -\nabla_{\mathbf{z}} E_{\boldsymbol{\theta}}(\mathbf{z}, \lambda) \qquad \text{(With the gradient of an energy-based model)} \qquad (108)$$

$$= -\hat{\boldsymbol{\epsilon}}_{\boldsymbol{\theta}}(\mathbf{z}; \lambda)/\sigma_\lambda \qquad \text{(With a noise prediction model)} \qquad (109)$$

$$= -\sigma_\lambda^{-2}(\mathbf{z} - \alpha_\lambda \hat{\mathbf{x}}_{\boldsymbol{\theta}}(\mathbf{z}; \lambda)) \qquad \text{(With a data prediction model)} \qquad (110)$$

We can let a neural network output any of $\mathbf{s}_{\boldsymbol{\theta}}(\mathbf{z}_\lambda; \lambda)$, $\hat{\boldsymbol{\epsilon}}_{\boldsymbol{\theta}}$ or $\hat{\mathbf{x}}_{\boldsymbol{\theta}}$, and we can convert the variables to each other using the equalities above.

The chosen relationship between $\mathbf{z}_\lambda$, $\hat{\mathbf{x}}$, $\hat{\boldsymbol{\epsilon}}$ and $\mathbf{s}_{\boldsymbol{\theta}}(\mathbf{z}_\lambda; \lambda)$ above, are due to the following relationships between $\mathbf{z}_\lambda$, $\mathbf{x}$ and $\boldsymbol{\epsilon}$:

$$\mathbf{z}_\lambda = \alpha_\lambda \mathbf{x} + \sigma_\lambda \boldsymbol{\epsilon} \qquad (111)$$

$$\mathbf{x} = \alpha_\lambda^{-1}(\mathbf{z}_\lambda - \sigma_\lambda \boldsymbol{\epsilon}) \qquad (112)$$

$$\boldsymbol{\epsilon} = \sigma_\lambda^{-1}(\mathbf{z}_\lambda - \alpha_\lambda \mathbf{x}) \qquad (113)$$

And:

$$\nabla_{\mathbf{z}_\lambda} \log q(\mathbf{z}_\lambda|\mathbf{x}) = \nabla_{\mathbf{z}_\lambda} - ||\mathbf{z}_\lambda - \alpha_\lambda \mathbf{x}||_2^2/(2\sigma_\lambda^2) \qquad (114)$$

$$= -\sigma_\lambda^{-2}(\mathbf{z}_\lambda - \alpha_\lambda \mathbf{x}) \qquad (115)$$

$$= -\sigma_\lambda^{-2}(\alpha_\lambda \mathbf{x} + \sigma_\lambda \boldsymbol{\epsilon} - \alpha_\lambda \mathbf{x}) \qquad (116)$$

$$= -\boldsymbol{\epsilon}/\sigma_\lambda \qquad (117)$$

In addition, there's the $\mathbf{v}$-prediction parameterization ($\mathbf{v} := \alpha_\lambda \boldsymbol{\epsilon} - \sigma_\lambda \mathbf{x}$) explained in D.2, and the $\mathbf{o}$-prediction parameterization ($\mathbf{o} := \mathbf{x} - \boldsymbol{\epsilon}$) explained in D.3.

Karras et al. [2022] proposed a specific $\mathbf{F}$-parametrization, with an $\mathbf{F}$-prediction model $\hat{\mathbf{F}}_{\boldsymbol{\theta}}$. In the special case of variance explosion (VE) SDE, it is formulated as:

$$\mathbf{x} = \frac{\tilde{\sigma}_{\text{data}}^2}{e^{-\lambda} + \tilde{\sigma}_{\text{data}}^2} \mathbf{z}_\lambda + \frac{e^{-\lambda/2} \tilde{\sigma}_{\text{data}}}{\sqrt{e^{-\lambda} + \tilde{\sigma}_{\text{data}}^2}} \mathbf{F} \qquad (118)$$

where $\tilde{\sigma}_{\text{data}} = 0.5$. Generalizing this to our more general family with arbitrary drift, this corresponds to:

$$\mathbf{x} = \frac{\tilde{\sigma}_{\text{data}}^2 \alpha_\lambda}{e^{-\lambda} + \tilde{\sigma}_{\text{data}}^2} \mathbf{z}_\lambda + \frac{e^{-\lambda/2} \tilde{\sigma}_{\text{data}}}{\sqrt{e^{-\lambda} + \tilde{\sigma}_{\text{data}}^2}} \mathbf{F} \qquad (119)$$

So that we have:

$$\mathbf{F} = \frac{\sqrt{e^{-\lambda} + \tilde{\sigma}_{\text{data}}^2}}{e^{-\lambda/2} \tilde{\sigma}_{\text{data}}} \mathbf{x} - \frac{\tilde{\sigma}_{\text{data}} \alpha_\lambda}{e^{-\lambda/2} \sqrt{e^{-\lambda} + \tilde{\sigma}_{\text{data}}^2}} \mathbf{z}_\lambda \qquad (120)$$

$$= -\frac{\sqrt{e^{-\lambda} + \tilde{\sigma}_{\text{data}}^2}}{\tilde{\sigma}_{\text{data}}} \boldsymbol{\epsilon} + \frac{e^{\lambda/2}(e^{-\lambda} + \tilde{\sigma}_{\text{data}}^2 - \tilde{\sigma}_{\text{data}}^2 \alpha_\lambda^2)}{\sqrt{e^{-\lambda} + \tilde{\sigma}_{\text{data}}^2} \tilde{\sigma}_{\text{data}} \alpha_\lambda} \mathbf{z}_\lambda \qquad (121)$$

In summary, given these different parameterizations, the $\boldsymbol{\epsilon}$-prediction loss can be written in terms of other parameterizations as follows:

$$||\boldsymbol{\epsilon} - \hat{\boldsymbol{\epsilon}}_{\boldsymbol{\theta}}||_2^2 = e^\lambda ||\mathbf{x} - \hat{\mathbf{x}}_{\boldsymbol{\theta}}||_2^2 \qquad (\boldsymbol{\epsilon}\text{-prediction and } \mathbf{x}\text{-prediction error}) \qquad (122)$$

$$= \sigma_\lambda^2 ||\nabla_{\mathbf{z}_\lambda} \log q(\mathbf{z}_\lambda|\mathbf{x}) - \mathbf{s}_{\boldsymbol{\theta}}||_2^2 \qquad \text{(score prediction)} \qquad (123)$$

$$= \alpha_\lambda^{-2}(e^{-\lambda} + 1)^{-2} ||\mathbf{v} - \hat{\mathbf{v}}_{\boldsymbol{\theta}}||_2^2 \qquad (\mathbf{v}\text{-prediction, general}) \qquad (124)$$

$$= (e^{-\lambda} + 1)^{-1} ||\mathbf{v} - \hat{\mathbf{v}}_{\boldsymbol{\theta}}||_2^2 \qquad (\mathbf{v}\text{-prediction with VP SDE}) \qquad (125)$$

$$= (e^{-\lambda}/\tilde{\sigma}_{\text{data}}^2 + 1)^{-1} ||\mathbf{F} - \hat{\mathbf{F}}_{\boldsymbol{\theta}}||_2^2 \qquad (\mathbf{F}\text{-prediction}) \qquad (126)$$

Interestingly, if we set $\tilde{\sigma}_{\text{data}}^2 = 1$, the training objectives of $\mathbf{F}$-prediction and $\mathbf{v}$-prediction are the same.

| Noise schedule name | $\lambda = f_\lambda(t) = ...$ | $t = f_\lambda^{-1}(\lambda) = ...$ | $p(\lambda) = -\frac{d}{d\lambda} f_\lambda^{-1}(\lambda) = ...$ |
|---|---|---|---|
| Cosine | $-2\log(\tan(\pi t/2))$ | $(2/\pi)\arctan(e^{-\lambda/2})$ | $\mathrm{sech}(\lambda/2)/(2\pi)$ |
| Shifted cosine | $-2\log(\tan(\pi t/2)) + 2s$ | $(2/\pi)\arctan(e^{-\lambda/2-s})$ | $\mathrm{sech}(\lambda/2 - s)/(2\pi)$ |
| EDM (training) | $-F_\mathcal{N}^{-1}(t; 2.4, 2.4^2)$ | $F_\mathcal{N}(-\lambda; 2.4, 2.4^2)$ | $\mathcal{N}(\lambda; 2.4, 2.4^2)$ |
| EDM (sampling) | $-2\rho\log(\sigma_{\max}^{1/\rho} + (1-t)(\sigma_{\min}^{1/\rho} - \sigma_{\max}^{1/\rho}))$ | $1 - \frac{e^{-\lambda/(2\rho)} - \sigma_{\max}^{1/\rho}}{\sigma_{\min}^{1/\rho} - \sigma_{\max}^{1/\rho}}$ | $\frac{e^{-\lambda/(2\rho)}}{2\rho(\sigma_{\max}^{1/\rho} - \sigma_{\min}^{1/\rho})}$ |
| Flow Matching with OT (see D.3) | $2\log((1-t)/t)$ | $1/(1 + e^{\lambda/2})$ | $\mathrm{sech}^2(\lambda/4)/8$ |

Table 4: Noise schedules used in our experiments: cosine [Nichol and Dhariwal, 2021], shifted cosine [Hoogeboom et al., 2023], and EDM [Karras et al., 2022] training and sampling schedules. Note that these are the noise schedules *before* truncation (Section E.3.1).

## E.3 Noise schedules

During model training, we sample time $t$ uniformly: $t \sim \mathcal{U}(0,1)$, then compute $\lambda = f_\lambda(t)$. This results in a distribution over noise levels $p(\lambda)$, whose cumulative density function (CDF) is given by $1 - f_\lambda^{-1}(\lambda)$. For $\lambda \in [\lambda_{\min}, \lambda_{\max}]$ the probability density function (PDF) is the derivative of the CDF, which is $p(\lambda_t) = -(d/d\lambda) f_\lambda^{-1}(\lambda) = -dt/d\lambda = -1/f_\lambda'(t)$. Outside of $[\lambda_{\min}, \lambda_{\max}]$ the probability density is 0.

In table 4 we provide some popular noise schedules: cosine [Nichol and Dhariwal, 2021], shifted cosine [Hoogeboom et al., 2023], and EDM [Karras et al., 2022]. We do not list the 'linear' schedule by [Ho et al., 2020], $f_\lambda(t) = -\log(e^{t^2} - 1)$ which has fallen out of use. Note that:

- In the shifted cosine schedule, $s = \log(64/d)$, where 64 is the base resolution and $d$ is the used resolution (e.g. 128, 256, 512, etc.).
- In the EDM training schedule, the function $F_\mathcal{N}(\lambda; \mu, \sigma^2)$ is the Normal distribution CDF, and $\mathcal{N}(\lambda; \mu, \sigma^2)$ is its PDF.
- In the EDM sampling schedule, $\rho = 7$, $\sigma_{\min} = 0.002$, $\sigma_{\max} = 80$. The density function $p(\lambda)$ in the table has support $\lambda \in [-\log\sigma_{\max}^2, -\log\sigma_{\min}^2]$. Outside this range, $p(\lambda) = 0$.

Note that the table gives the noise schedules *before* truncation, resulting in $[\widetilde{f}_\lambda(0), \widetilde{f}_\lambda(1)] = [\infty, -\infty]$. The truncation procedure is given in E.3.1.

### E.3.1 Truncation

The noise schedules above are truncated, resulting in a noise schedule $\widetilde{f}_\lambda(0)$ whose endpoints have desired values $[\widetilde{f}_\lambda(0), \widetilde{f}_\lambda(1)] = [\lambda_{\max}, \lambda_{\min}]$:

$$\widetilde{f}_\lambda(t) := f_\lambda(t_0 + (t_1 - t_0)t) \tag{127}$$

$$\text{where: } t_0 := f_\lambda^{-1}(\lambda_{\max}) \tag{128}$$

$$t_1 := f_\lambda^{-1}(\lambda_{\min}) \tag{129}$$

Its inverse is:

$$\widetilde{f}_\lambda^{-1}(\lambda) = (f_\lambda^{-1}(\lambda) - t_0)/(t_1 - t_0) \tag{130}$$

And the corresponding probability density:

$$\text{if } \lambda_{\min} \leq \lambda \leq \lambda_{\max}: \quad \tilde{p}(\lambda) = -\frac{d}{d\lambda}\widetilde{f}_\lambda^{-1}(\lambda) = -\frac{d}{d\lambda}f_\lambda^{-1}(\lambda)/(t_1 - t_0) = p(\lambda)/(t_1 - t_0) \tag{131}$$

$$\text{else: } \tilde{p}(\lambda) = 0 \tag{132}$$

## E.4 Sampling

Anderson [1982] showed that if $\mathbf{s}_\theta(\mathbf{z}; \lambda) = \nabla_\mathbf{z} \log q_t(\mathbf{z})$, then the forward SDE is exactly reversed by the following SDE:

$$d\mathbf{z} = [\mathbf{f}(\mathbf{z}, t) - g(t)^2 \mathbf{s}_\theta(\mathbf{z}; \lambda)]dt + g(t)d\mathbf{w} \tag{133}$$

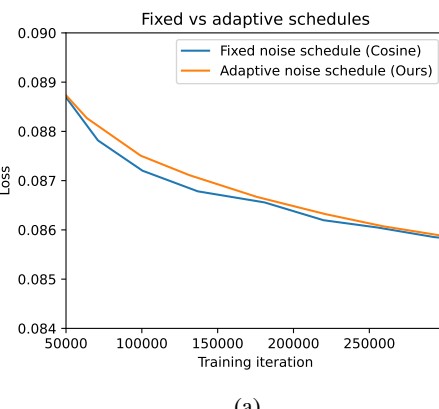
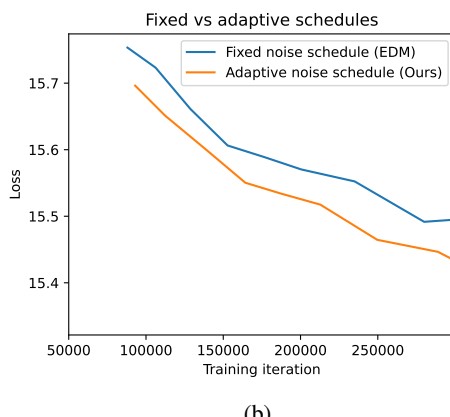

| (a) | (b) |

Figure 4: Our proposed adaptive noise schedule allows us to freely change the weighting function without needing to handtune a corresponding noise schedule. For some models it leads to slightly worse convergence speed (left) compared to a static noise schedule, probably because the noise schedule was already well-tuned for the weighting function, while in other cases it led to faster convergence (right). The adaptive noise schedule did not significantly affect the end result, but it did allow us to more freely experiment with weighting functions.

Recent diffusion models have used increasingly sophisticated samplers. As an alternative to solving the SDE, [Song et al., 2021b] showed that sampling from the model can alternatively be done by solving the following *probability flow* ODE:

$$d\mathbf{z} = [\mathbf{f}(\mathbf{z}, t) - \frac{1}{2}g(t)^2 \mathbf{s}_{\boldsymbol{\theta}}(\mathbf{z}; \lambda)]dt \tag{134}$$

which, under the assumption that $\mathbf{s}_{\boldsymbol{\theta}}$ is a conservative vector field, will result in the same marginals $p(\mathbf{z}_t)$ as the SDE of Equation 133 for every $t \in [0, 1]$, and therefore also the same marginal $p(\mathbf{x})$.

Note that due to the continuous-time nature of the model, any sampling method is necessarily approximate, with the discretization error depending on various factors including the choice of noise schedule. For sampling we can therefore typically use a different noise schedule $f_\lambda$ for sampling than for training, and we can change the SDE drift term; as long as we appropriately rescale the input to the score network, this would still result in correct samples.

## F  Adaptive noise schedule

The invariance shown in Section 3.2 holds for the loss $\mathcal{L}_w(\mathbf{x})$, but not for the Monte Carlo estimator of the loss that we use in training, based on random samples $t \sim \mathcal{U}(0, 1), \epsilon \sim \mathcal{N}(0, \mathbf{I})$. The noise schedule still affects the *variance* of this Monte Carlo estimator and its gradients; therefore, the noise schedule affects the efficiency of optimization.

In fact, the noise schedule acts as an importance sampling distribution for estimating the loss integral of Equation 6. Specifically, note that $p(\lambda) = -1/(d\lambda/dt)$. We can therefore rewrite the weighted loss as the following, which clarifies the role of $p(\lambda)$ as an importance sampling distribution:

$$\mathcal{L}_w(\mathbf{x}) = \frac{1}{2}\mathbb{E}_{\epsilon \sim \mathcal{N}(0,\mathbf{I}), \lambda \sim p(\lambda)}\left[\frac{w(\lambda)}{p(\lambda)}||\hat{\boldsymbol{\epsilon}}_{\boldsymbol{\theta}}(\mathbf{z}_\lambda; \lambda) - \epsilon||_2^2\right] \tag{135}$$

In order to avoid having to hand-tune the noise schedule for different weighting functions, we implemented an adaptive noise schedule. The noise schedule $\lambda_t$ is updated online, where we let $p(\lambda) \propto \mathbb{E}_{\mathbf{x} \sim \mathcal{D}, \epsilon \sim \mathcal{N}(0,\mathbf{I})}[w(\lambda)||\hat{\boldsymbol{\epsilon}}_{\boldsymbol{\theta}}(\mathbf{z}_\lambda; \lambda) - \epsilon||_2^2]$. This noise schedule ensures that the loss is spread evenly over time, i.e. that the magnitude of the loss $\mathbb{E}_{\mathbf{x} \sim \mathcal{D}, \epsilon \sim \mathcal{N}(0,\mathbf{I})}\left[(w(\lambda)/p(\lambda))||\hat{\boldsymbol{\epsilon}}_{\boldsymbol{\theta}}(\mathbf{z}_\lambda; \lambda) - \epsilon||_2^2\right]$ is approximately invariant to $\lambda$ or $t$. We find that this often significantly speeds op optimization.

We implemented an adaptive noise schedule $p(\lambda)$, where:

$$p(\lambda) \propto \mathbb{E}_{\mathbf{x} \sim \mathcal{D}, \epsilon \sim \mathcal{N}(0,\mathbf{I})}[w(\lambda)||\epsilon - \hat{\boldsymbol{\epsilon}}_{\boldsymbol{\theta}}(\mathbf{z}_\lambda; \lambda)||_2^2] \tag{136}$$

In practice we approximate this by dividing the range $[\lambda_{\min}, \lambda_{\max}]$ into 100 evenly spaced bins, and during training keep an exponential moving average (EMA) of $w(\lambda)||\epsilon - \hat{\epsilon}_{\boldsymbol{\theta}}(\mathbf{z}_\lambda; \lambda)||_2^2$ within each bin. From these EMAs we construct a piecewise linear function $f_\lambda(t)$ such that Equation 136 is approximately satisfied. The EMAs and corresponding noise schedule $p(\lambda)$ are updated at each training iteration. Similar noise schedules have been proposed in Zheng et al. [2023], Nichol and Dhariwal [2021], Dieleman et al. [2022].

In experiments we measure the effect of changing the fixed noise schedule of existing modules with an adaptive schedule. We found that this lead to approximately equal FID scores. In half of the experiments, optimization was approximately as fast as with the original noise schedule, while in the other half the adaptive noise schedule lead to faster optimization (see Figure 4). The end results were not significantly altered.

## G Relationship between the KL divergence and Fisher divergence

We'll use the following definition of the Fisher divergence [Lyu, 2012]:

$$D_F(q(\mathbf{x})||p(\mathbf{x})) := \mathbb{E}_q(\mathbf{x})[||\nabla_{\mathbf{x}} \log q(\mathbf{x}) - \nabla_{\mathbf{x}} \log p(\mathbf{x})||_2^2] \tag{137}$$

**Theorem 2.** *Assume a model in the family specified in Section 2, and assume the score network encodes a conservative vector field: $\mathbf{s}_{\boldsymbol{\theta}}(\mathbf{z}_t, \lambda_t) = \nabla_{\mathbf{z}_t} \log p(\mathbf{z}_t)$ (not assumed by the other theorems). Then:*

$$\frac{d}{d\lambda} D_{KL}(q(\mathbf{z}_{t,\dots,1}|\mathbf{x})||p(\mathbf{z}_{t,\dots,1})) = \frac{1}{2}\sigma_\lambda^2 D_F(q(\mathbf{z}_t|\mathbf{x})||p(\mathbf{z}_t)) \tag{138}$$

*Proof of Theorem 2.* Note that (see Equation 117):

$$\nabla_{\mathbf{z}_t} \log q(\mathbf{z}_t|\mathbf{x}) = -\boldsymbol{\epsilon}/\sigma_\lambda \tag{139}$$

And assume the score network encodes a conservative vector field:

$$\nabla_{\mathbf{z}_t} \log p(\mathbf{z}_t) = \mathbf{s}_{\boldsymbol{\theta}}(\mathbf{z}_t, \lambda_t) = -\hat{\boldsymbol{\epsilon}}_{\boldsymbol{\theta}}(\mathbf{z}_t; t)/\sigma_\lambda \tag{140}$$

So the time derivative of Equation 22 can be expressed as:

$$\frac{d}{d\lambda} D_{KL}(q(\mathbf{z}_{t,\dots,1}|\mathbf{x})||p(\mathbf{z}_{t,\dots,1})) = \frac{1}{2}\sigma_\lambda^2 \mathbb{E}_{q(\mathbf{z}_t|\mathbf{x})}\left[||\nabla_{\mathbf{z}_t} \log q(\mathbf{z}_t|\mathbf{x}) - \nabla_{\mathbf{z}_t} \log p(\mathbf{z}_t)||_2^2\right] \tag{141}$$

Equation 138 follows from the definition of the Fisher divergence. ∎

### G.1 Comparison with Theorem 1 by Lyu [2012]

Lyu [2012] prove a similar result in their Theorem 1. We'll translate their result into our notation. In particular, let the forward process be as in our family, such that $q(\mathbf{z}_t|\mathbf{x}) = \mathcal{N}(\mathbf{z}_t; \alpha_\lambda \mathbf{x}, \sigma_t^2 \mathbf{I})$. The marginal (data) distribution is $q(\mathbf{x})$, such that $q(\mathbf{z}_t) = \int q(\mathbf{z}_t|\mathbf{x})q(\mathbf{x})d\mathbf{x}$. Similarly, let the generative model have a marginal $p(\mathbf{x})$, and $p(\mathbf{z}_t) = \int p(\mathbf{z}_t|\mathbf{x})p(\mathbf{x})d\mathbf{x}$. So far the assumptions are the same in our family.

They assume that $q(\mathbf{z}_t|\mathbf{x}) = \mathcal{N}(\mathbf{z}_t, \mathbf{x}, t)$, which corresponds to a variance exploding (VE) diffusion process, with $t = \sigma_\lambda^2$, so $\lambda = -\log(t)$. Importantly, they make the assumption that $p(\mathbf{z}_t|\mathbf{x}) = q(\mathbf{z}_t|\mathbf{x})$, i.e. that the forward process for $p$ equals the forward process for $q$. Given these assumptions, Lyu [2012] show that:

$$\frac{d}{dt} D_{KL}(q(\mathbf{z}_t)||p(\mathbf{z}_t)) = -\frac{1}{2} D_F(q(\mathbf{z}_t)||p(\mathbf{z}_t)) \tag{142}$$

Which, given the noise schedule $\lambda = -\log(t)$, can be rewritten as:

$$\frac{d}{d\lambda} D_{KL}(q(\mathbf{z}_t)||p(\mathbf{z}_t)) = \frac{1}{2}\sigma_\lambda^2 D_F(q(\mathbf{z}_t)||p(\mathbf{z}_t)) \tag{143}$$

which looks a lot like our Equation 138. One difference are that in Equation 138, the left-hand-side distributions are joint distributions, and $q$ conditions on $\mathbf{x}$, while Equation 143 is about the

unconditional $q$. Another key difference is that for Equation 138 we need fewer assumptions: most importantly, we do not make the assumption that $p(\mathbf{z}_t|\mathbf{x}) = q(\mathbf{z}_t|\mathbf{x})$, since this assumption does *not* hold for the family of diffusion models we consider. Before or during optimization, $p(\mathbf{z}_t|\mathbf{x})$ might be very far from $q(\mathbf{z}_t|\mathbf{x})$. After optimization, $p(\mathbf{z}_t|\mathbf{x})$ might be close to $q(\mathbf{z}_t|\mathbf{x})$, but we still can't assume they're equal. In addition, we're mostly interested in the properties of the loss function during optimization, since that's when we're using our loss for optimization. We for this reason, our Theorem 2 is a lot more relevant for optimization.

## H   Implementation details

Instead of uniformly sampling $t$, we applied the low-discrepency sampler of time that was proposed by Kingma et al. [2021], which has been shown to effectively reduce the variance of diffusion loss estimator and lead to faster optimization. The model is optimized by *Adam* [Kingma and Ba, 2014] with the default hyperparameter settings. We clipped the learning gradient with a global norm of 1.

For the adaptive noise schedules, we divided the range of $[\lambda_{\min}, \lambda_{\max}]$ into 100 evenly spaced bins. During training, we maintaiedn an exponential moving average of $w(\lambda)\|\boldsymbol{\epsilon} - \hat{\boldsymbol{\epsilon}}_{\boldsymbol{\theta}}(\mathbf{z}_\lambda; \lambda)\|_2^2$ with a decay rate 0.999, and a constant initialization value of 1 for each bin.

Below we elaborate the implementation details specific for each task.

**ImageNet 64x64.**   For class-conditional generation on ImageNet 64x64, we applied the ADM U-Net architecture from Dhariwal and Nichol [2022], with dropout rate 0.1. We didn't use any data augmentation. The model was trained with learning rate $1e-4$, exponential moving average of 50 million images and learning rate warmup of 10 million images, which mainly follows the configuration of Karras et al. [2022]. We employed 128 TPU-v4 chips with a batch size of 4096 (32 per chip). We trained the model for 700k iterations and reported the performance of the checkpoint giving the best FID score (checkpoints were saved and evaluated on every 20k iterations). It took around 3 days for a single training run. For training noise schedule and sampling noise schedule of DDPM sampler, we set $\lambda_{\min} = -20$ and $\lambda_{\max} = 20$. We fixed the noise schedule used in sampling to the cosine schedule for the DDPM sampler, and the EDM (sampling) schedule for the EDM sampler (see Table 4 for the formulations). We adopted the same hyperparameters of EDM sampler from Karras et al. [2022] with no changes (i.e., Table 5 in their work, column 'ImageNet-Our model'). Both DDPM and EDM samplers took 256 sampling steps.

**ImageNet 128x128.**   For class-conditional generation on ImageNet 128x128, we heavily followed the setting of *simple diffusion* [Hoogeboom et al., 2023]. Specifically, we used their 'U-ViT, L' architecture, and followed their learning rate and EMA schedules. The data was augmented with random horizontal flip. The model is trained using 128 TPU-v4 chips with a batch size of 2048 (16 per chip). We trained the model for 700 iterations and evaluated the FID and inception scores every 100k iterations. The results were reported with the checkpoint giving the best FID score. It took around 7 days for a single run. We set $\lambda_{\min} = -15 + s$ and $\lambda_{\max} = 15 + s$, where $s = \log(64/d)$ is the shift of the weighting function, with 64 being the base resolution and $d$ being the model resolution ($d = 128$ for this task). The DDPM sampler used for evaluation used 'shifted-cosine' noise schedule (Table 4) and took 512 sampling steps.

## I   Relationship with low-bit training

Various earlier work, such as [Kingma and Dhariwal, 2018], found that maximum likelihood training on 5-bit data can lead to perceptually higher visual quality than training on 8-bit data (at the cost of a decrease in color fidelity). This leads to improved visual quality, probably because it allows the model to spend more capacity on modeling the bits that are most relevant for human perception.

In [Kingma and Dhariwal, 2018], training on 5-bit data was performed by adding uniform noise to the data, before feeding it to the model. It was found that adding Gaussian noise had a similar effect as uniform noise. As we have seen, in the case of diffusion models, adding Gaussian noise is equivalent to using a weighted objective with a monotonic weighting function.

Therefore, training on 5-bit data is similar to training using a monotonic weighting function in case of diffusion models. We can wonder: which weighting function emulates training on 5-bit data? Here, we'll attempt to answer this question.

## I.1 The shape of $\frac{d}{d\lambda}\mathcal{L}(\lambda; \mathbf{x})$ for low-bit data

Note that the results of Appendix A.1 can also be written as:

$$\frac{d}{d\lambda}\mathcal{L}(\lambda; \mathbf{x}) = \frac{1}{2}\mathbb{E}_{\boldsymbol{\epsilon}\sim\mathcal{N}(0,\mathbf{I})}\left[||\boldsymbol{\epsilon} - \hat{\boldsymbol{\epsilon}}_{\boldsymbol{\theta}}(\mathbf{z}_\lambda; \lambda)||_2^2\right] \tag{144}$$

This allows us to rewrite the weighted loss as simply:

$$\mathcal{L}_w(\mathbf{x}) = \int_{\lambda_{\max}}^{\lambda_{\min}} \frac{d}{d\lambda}\mathcal{L}(\lambda; \mathbf{x})\, w(\lambda)\, d\lambda \tag{145}$$

To understand the effect of $\lambda$, we can plot the $\frac{d}{d\lambda}\mathcal{L}(\lambda; \mathbf{x})$ as a function of $\lambda$.

We'd like to plot $\frac{d}{d\lambda}\mathcal{L}(\lambda; \mathbf{x})$ dof different choices of bit precision. This will tell us where the different bits 'live' as a function of $\lambda$. Since training different diffusion models on different bit precisions is very expensive, we instead use an approximation. In particular, we assume that the data $\mathbf{x}$ is univariate, with a uniform distribution $q(\mathbf{x})$ over the $2^n$ possible pixel values, where $n$ is the bit precision. The data is $\mathbf{x}$ is, as usual, normalized to $[-1, 1]$. We then let the model $p(\mathbf{z}_\lambda)$, for each choice of $\lambda$, be the optimal model: $p(\mathbf{z}_\lambda) := \int q(\mathbf{x})q(\mathbf{z}_\lambda|\mathbf{x})d\mathbf{z}$, which is a univariate mixture-of-Gaussians, where each mixture component is a Gaussian centered one of the $2^n$ possible pixel values. In this case, $\mathcal{L}(\lambda; \mathbf{x}) := D_{KL}(q(\mathbf{z}_\lambda|\mathbf{x})||p(\mathbf{z}_\lambda))$.

We start by plotting the function $\mathbb{E}_{q(\mathbf{x})}[\mathcal{L}(\lambda; \mathbf{x})]$, for differences choices of $n = 1, ..., 8$, as shown in Figure 5a. Next, we plot $\mathbb{E}_{q(\mathbf{x})}[\frac{d}{d\lambda}\mathcal{L}(\lambda; \mathbf{x})]$, for each $n$, in Figure 5b. The more bits included in the data, the more information contained by $\mathbb{E}_{q(\mathbf{x})}[\frac{d}{d\lambda}\mathcal{L}(\lambda; \mathbf{x})]$ (measured by the area under each curve). We can further visualize the contribution of each additional added bit to the loss, by subtracting the curve for $n-1$ bits from the curve for $n$ bits, as displayed in Figure 5c. The area under each of these curves is exactly 1 bit. Interestingly, as more bits being added, the contribution to the loss gradually shifts from low-SNR regions to high-SNR regions.

Now the connection between low-bit training (e.g., 5-bit) and the weighted training objective becomes clear: If we want to "shut down" the last three curves in Figure 5c, a way is to have a sigmoidal weighting function that goes down sharply between $\lambda = 7.5$ and $\lambda = 10$. In fact, the 5-bit unweighted loss curve is very similar to the 8-bit loss curve, when using the following weighting function:

$$w(\lambda) = F_{\mathcal{N}}((-2(\lambda - 8.4))) \tag{146}$$

where $F_{\mathcal{N}}$ is the CDF of a standard Gaussian. See Figure 5e for the comparison of the 5-bit unweighted loss and the 8-bit weighted loss, and Figure 5d for the visualization of the above weighting function ('5-bit-like weighting'). Interestingly, this weighting function gives much more weight to low noise levels than the weighting functions used in this paper (i.e. the other curves shown in Figure 5d).

## J  Fourier analysis of Gaussian noise perturbation

Recall Theorem 1 showed that assuming monotonic weightings, the weighted training objective is equivalent to the ELBO with data augmentation, namely Gaussian noise perturbation. One may wonder, why Gaussian noise perturbation can help improve perceptual quality. In this section, we try to answer this question by lifting Gaussian noise perturbation to the frequency domain.

Given a clean natural image $\mathbf{x}$ and a Gaussian white noise image $\boldsymbol{\epsilon}$, the Gaussian-noise-perturbed data is given by $\mathbf{z}_\lambda = \alpha_\lambda \mathbf{x} + \sigma_\lambda \boldsymbol{\epsilon}$. Denote the Fourier transform operation as $\mathcal{F}(\cdot)$. Since $\mathcal{F}(\cdot)$ is a linear operation, we have

$$\mathcal{F}(\mathbf{z}_\lambda) = \alpha_\lambda \mathcal{F}(\mathbf{x}) + \sigma_\lambda \mathcal{F}(\boldsymbol{\epsilon}). \tag{147}$$

To understand how noise affects the data, we can compare the power spectra of $\alpha_\lambda \mathcal{F}(\mathbf{x})$ and $\sigma_\lambda \mathcal{F}(\boldsymbol{\epsilon})$, which is defined as the square amplitude of the Fourier transform: $\mathcal{S}(\cdot) = |\mathcal{F}(\cdot)|^2$. More specifically, given the power spectrum $\mathcal{S}$ of an image, we can take the circular average of $\mathcal{S}$ for the Fourier

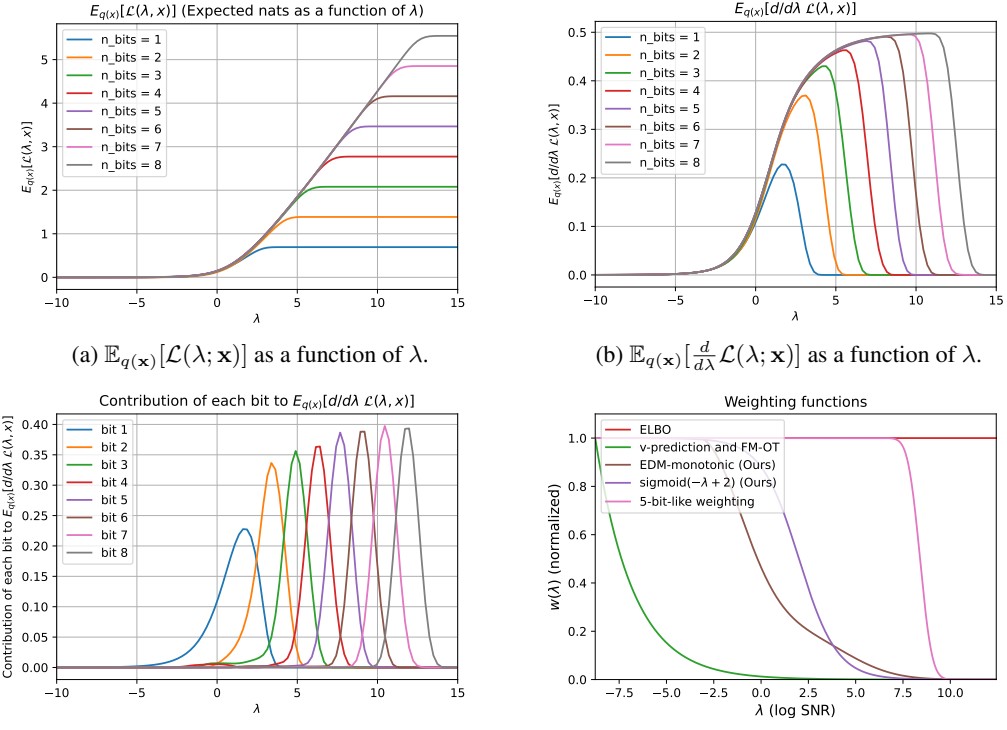

(a) $\mathbb{E}_{q(\mathbf{x})}[\mathcal{L}(\lambda; \mathbf{x})]$ as a function of $\lambda$.

(b) $\mathbb{E}_{q(\mathbf{x})}\left[\frac{d}{d\lambda}\mathcal{L}(\lambda; \mathbf{x})\right]$ as a function of $\lambda$.

(c) Subtraction of the curve for $n-1$ bits from the curve for $n$ bits in Fig.5b.

(d) Weighting functions, with a weighting resembling 5-bit training.

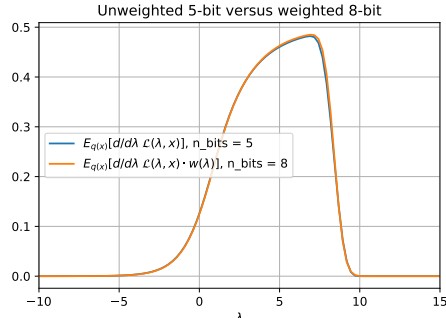

(e) Unweighted 5-bit loss versus weighted 8-bit loss.

Figure 5: Low-bit training analysis. See Section I.1 for description.

components of the same spatial frequency, such that it becomes a function of the spatial frequency. Figure 6 displays the average power spectra of $\alpha_\lambda \mathcal{F}(\mathbf{x})$ and $\sigma_\lambda \mathcal{F}(\boldsymbol{\epsilon})$ as functions of the spatial frequency, for different $\lambda$. As $\lambda$ goes from high to low, i.e., gradually perturbing the data with more noise, the average power spectrum of $\sigma_\lambda \mathcal{F}(\boldsymbol{\epsilon})$ (orange curve) first exceeds the one of $\alpha_\lambda \mathcal{F}(\mathbf{x})$ (blue curve) in the high frequency region, and then followed by the low frequency region. That is, the high frequency components of the data are destroyed more quickly than the low frequency components. When the data is perturbed with small amounts of noise, only high frequency details are significantly removed, and most low frequency content remains. Therefore, given that the model capacity is limited, learning data with Gaussian noise perturbation help the model focus more on low frequency components of the data, which often correspond to high level content and global structure that are more crucial to human perception. This analysis can also be applied to understand the forward process of diffusion models.

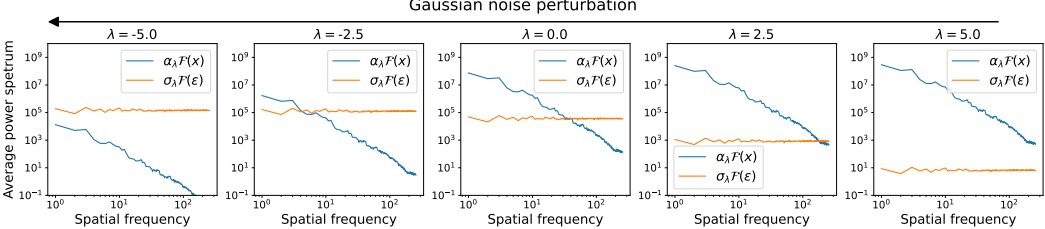

Figure 6: The circular average power spectrum as a function of spatial frequency, for the scaled clean image and scaled Gaussian noise image. From the rightmost to the leftmost subfigure, more noise is gradually added.

To make the claim even clearer, we can compute the log signal-to-noise-ratio (log SNR) $\tilde{\lambda}$ between the power spectra of $\alpha_\lambda \mathcal{F}(\mathbf{x})$ and $\sigma_\lambda \mathcal{F}(\boldsymbol{\epsilon})$, as a function of $\lambda$. Figure 7 shows the average value of that log ratio around different frequency regions. Given a fixed $\lambda$, $\tilde{\lambda}$ is higher for low frequency regions, so that the data are less destroyed by the noise.

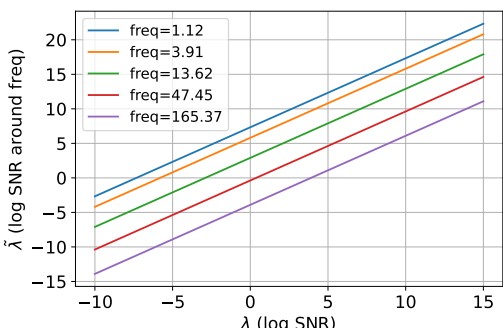

Figure 7: Log-SNR ($\tilde{\lambda}$) of the power spectra of the scaled clean image and scaled Gaussian noise image, as a function of log-SNR ($\lambda$) in the data space. Given a fixed $\lambda$, $\tilde{\lambda}$ is higher around low frequency regions.

## K   Limitations

It is important to emphasize that our empirical results, like other deep learning approaches, depend on the choice of hyper-parameters. A change in, for example, the dataset or spatial resolution will generally require re-tuning of optimization hyperparameters, architectural choices and/or weighting functions. Such re-tuning can be time consuming and costly.

## L   Broader impact

While our work primarily focuses on theoretical developments in the understanding and optimization of diffusion models, the advancements could have broader implications, some of which could potentially be negative. The development of more efficient and effective generative models could, on one hand, propel numerous beneficial applications, such as art and entertainment. However, it is also worth acknowledging the potential misuse of these technologies.

One notable concern is the generation of synthetic media content, for example to mislead. These fraudulent yet realistic-looking images and videos could be used for spreading disinformation or for other malicious purposes, such as identity theft or blackmail.

Regarding fairness considerations, generative models are typically trained on large datasets and could therefore inherit and reproduce any biases present in the training data. This could potentially result

in unfair outcomes or perpetuate harmful stereotypes if these models are used in decision-making processes or content generation.

Mitigation strategies to address these concerns could include the gated release of models, where access to the model or its outputs is regulated to prevent misuse. Additionally, the provision of defenses, such as methods to detect AI generated media, could be included alongside the development of the generative models. Monitoring mechanisms could also be implemented to observe how a system is being used and to ensure that it learns from feedback over time in an ethical manner.

These issues should be considered when applying the techniques we propose. The community should strive for an open and ongoing discussion about the ethics of AI and the development of strategies to mitigate potential misuse of these powerful technologies.

## M    Samples from our model trained on 512 × 512 ImageNet

Below we provide random samples from our highest-resolution (512x512) model trained on ImageNet. We did not cherry-pick, except that we removed depictions of humans due to ethical guidelines. Samples in Figures 8 and 9 are generated without guidance, while samples in Figures 10 and 11 are generated with guidance strength 4.

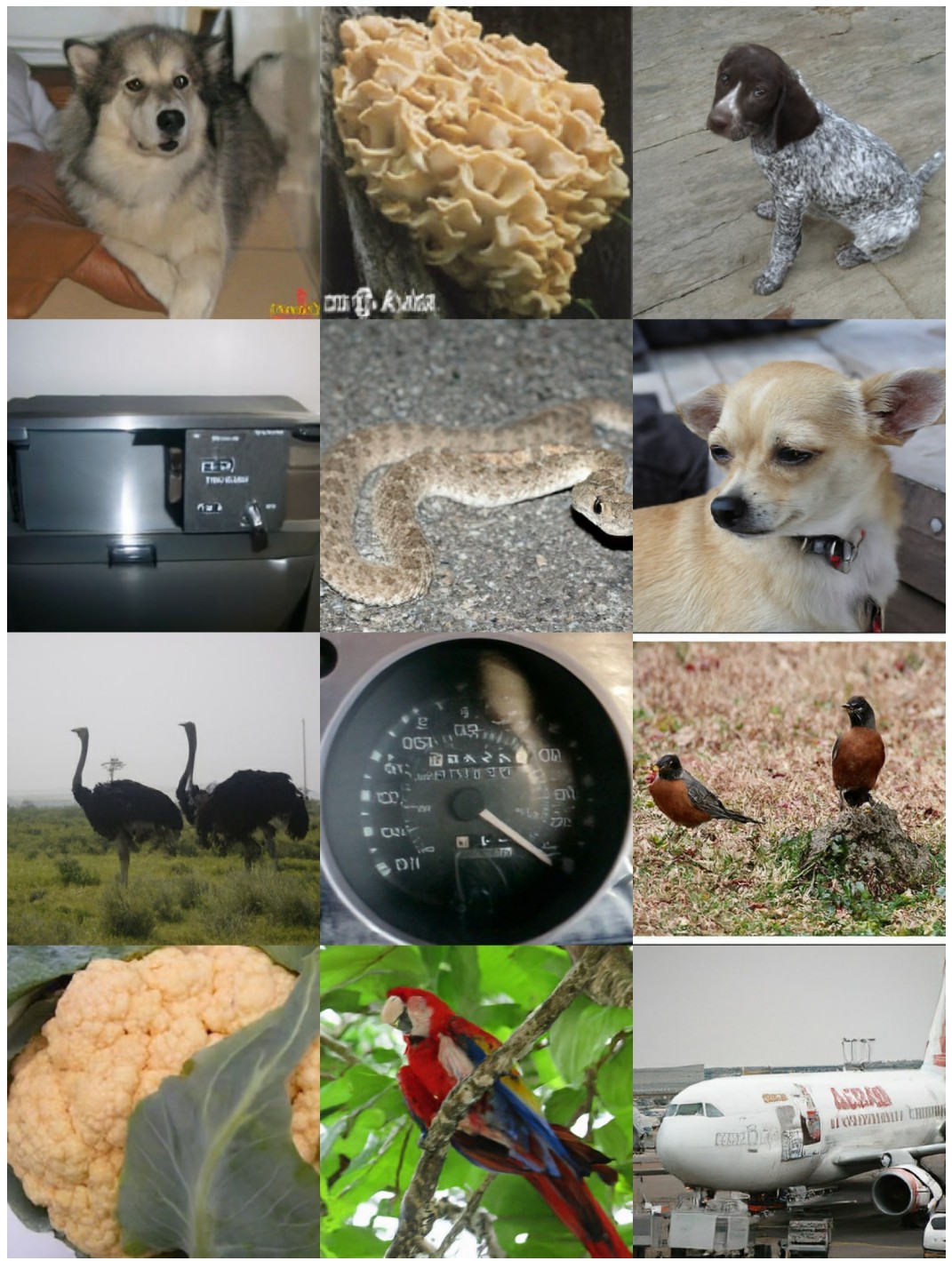

Figure 8: Random samples from our 512x512 ImageNet model, without guidance.

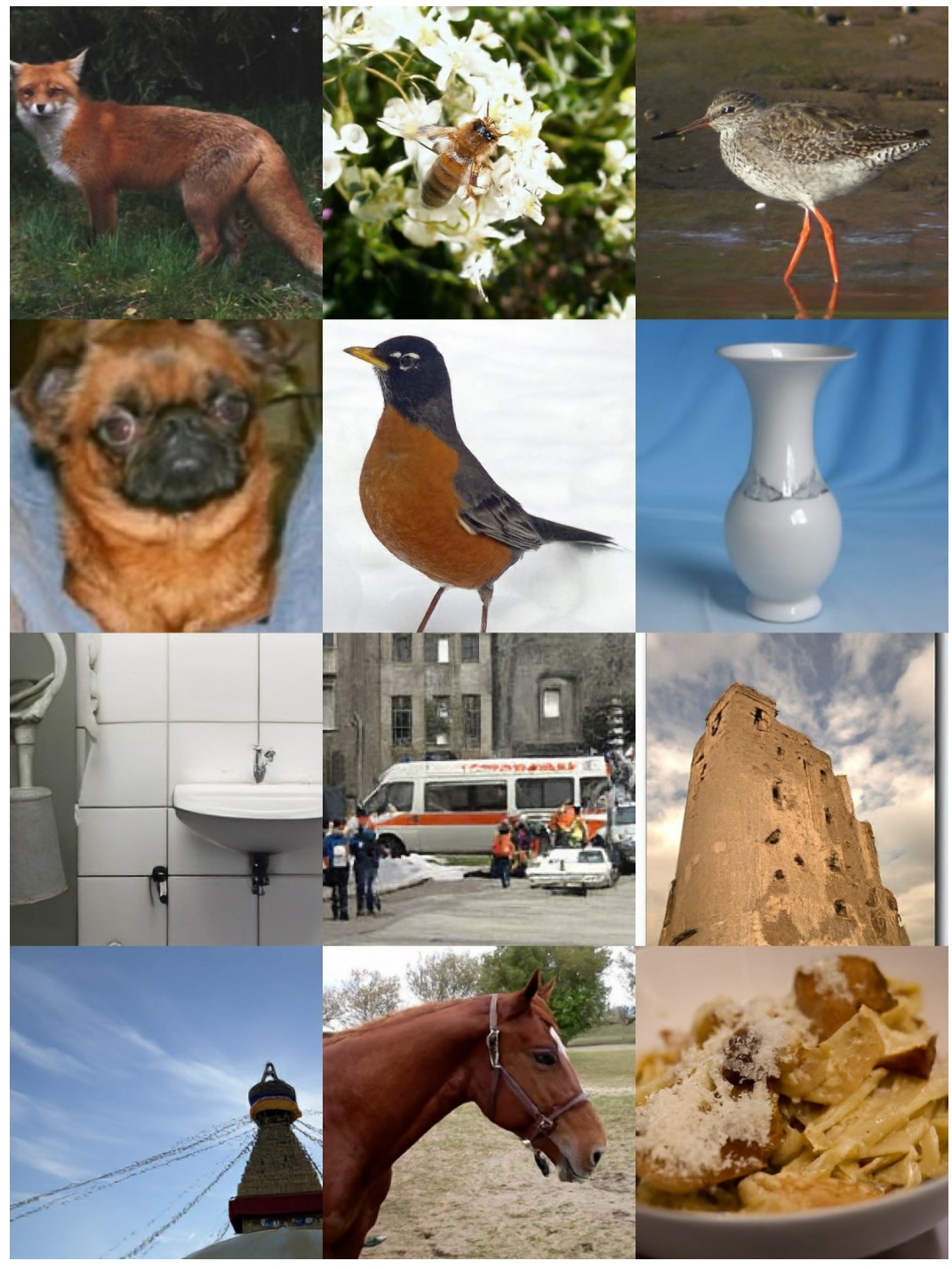

Figure 9: More random samples from our 512x512 ImageNet model, without guidance.

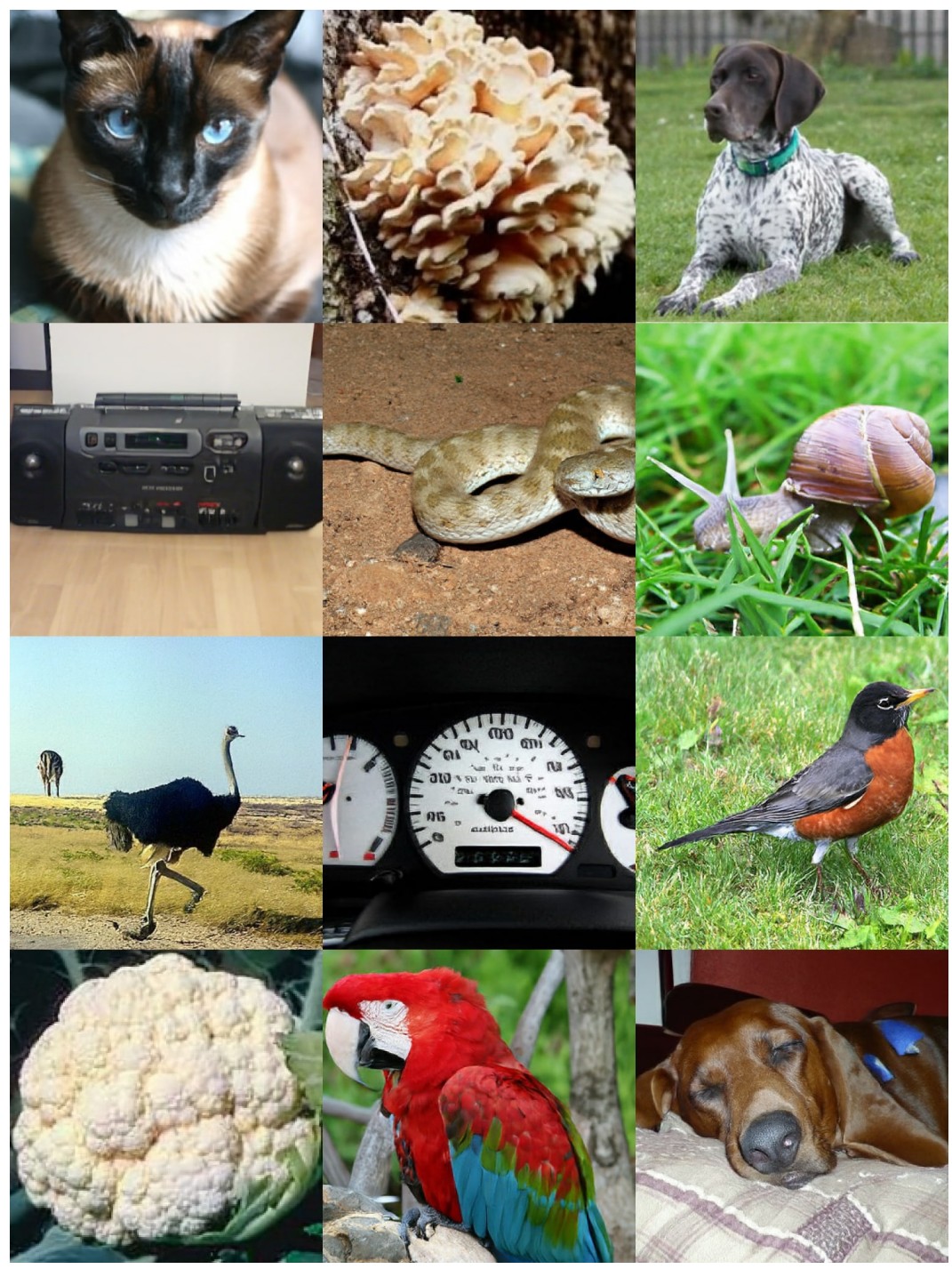

Figure 10: Random samples from our 512x512 ImageNet model, with guidance strength 4.

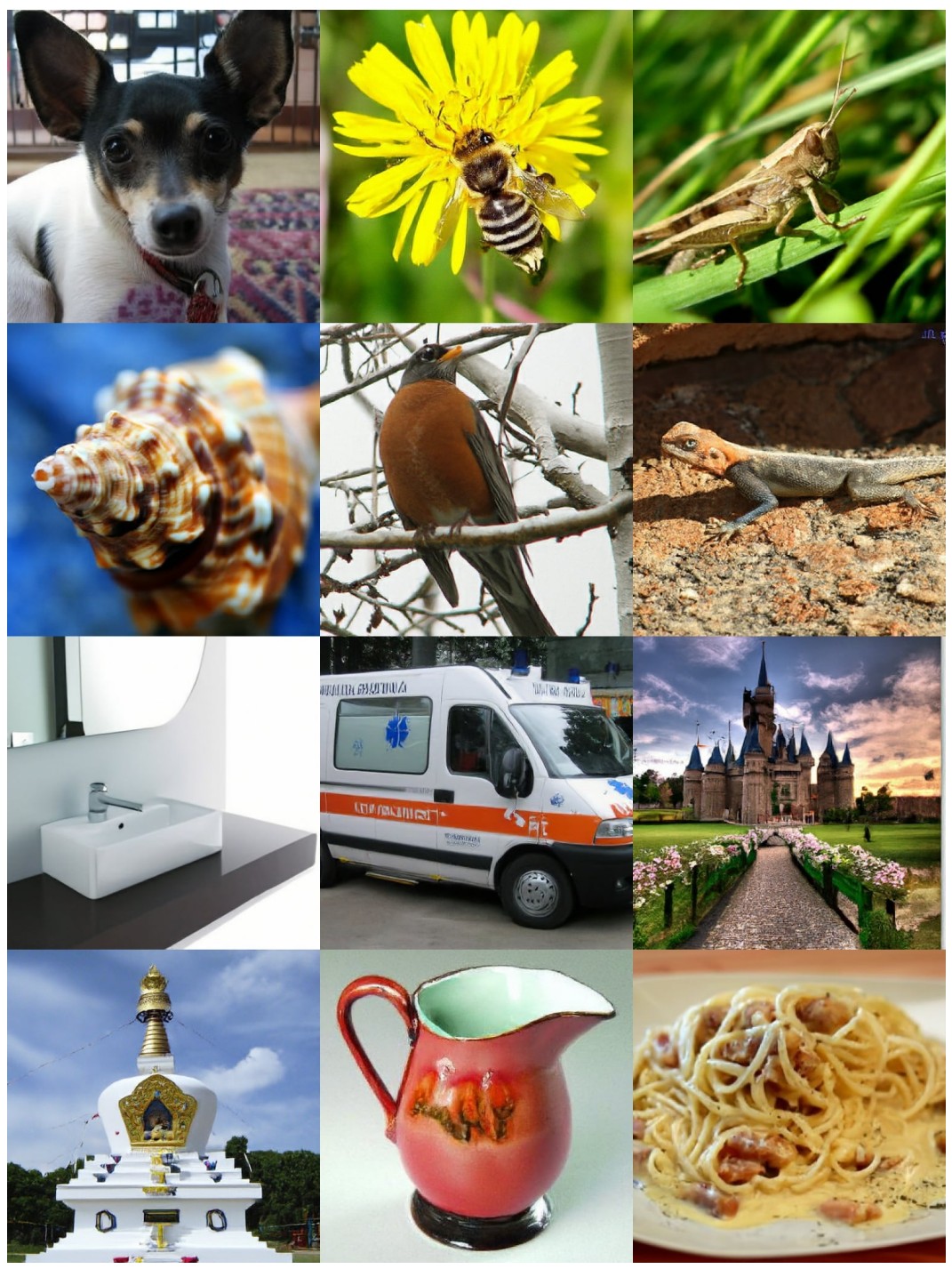

Figure 11: More random samples from our 512x512 ImageNet model, with guidance strength 4.

