# OpenReview forum: "Understanding Diffusion Objectives as the ELBO with Simple Data Augmentation"
_NeurIPS.cc/2023/Conference — NeurIPS 2023 oral_

### Official Review · Reviewer_Qo9S · 2023-07-05

**Soundness:** 3 good
**Presentation:** 2 fair
**Contribution:** 3 good
**Rating:** 6
**Confidence:** 4

**Summary:**

The paper studies different weighting schemes of the loss functions of generative diffusion models. In particular, the authors investigate the relationship between a weighted (with respect to noise levels) loss and the proper Evidence Lower Bound (ELBO), discussing how the former can be written as a weighted integral of ELBOs.

It is further claimed that when the weighting factor is monotonic, there exists an interpretation of the loss as the result of log-likelihood optimization considering data augmentation, in particular noise perturbation.

Finally, the proposed interpretation is leveraged to construct new weighting schemes, corroborating the discussion with empirical validation.


**Strengths:**

* Deepening our understanding of diffusion models is an important endeavour, given the popularity of such methods, and this paper is aligned with research lines on methodological contributions on training such models

* The authors attempted at reproducing the work from Karras et al, and provide an additional insights on the difficulty of obtaining generative performance that is on par with prior work

**Weaknesses:**

On a technical level, the paper has several problems related to mathematical rigor. Overall, a colloquial exposition is preferred over a clear mathematical language.

* Starting with line 76, the "series of latent variables" $z_{0,...,1}$ is introduced without specifying whether the set of time indices is uncountable (i.e. a continuous set between 0 and 1) or whether it is the collection of a finite set of time indices (for example an equally spaced grid between 0 and 1). The term "conditional distribution" $q(z_{0,...,1} | x)$ introduced at line 77 is very confusing, and requires a proper discussion, in particular for the case of uncountable time indices.

* This is not just a minor stylistic note, but has concrete impact on the derivation and in the discussions: for example, how is the KL divergence introduced at line 126, $D_{\text{KL}}\left(  q(z_{0,..1}),p(z_{0,..1}) \right)$, defined? Are the authors talking about path measures? In this case, a proper characterization needs the formal tools of stochastic calculus, like for example in Huang2021, where the Girsanov Theorem is used to construct the ELBO.

* Furthermore, in the different derivations I find confusing and counterproductive to switch back and forth between different notations (for example dropping or not the dependency on $t$ of $\lambda$). Why for example in equation (5) would you use lambda both with and without $t$?

* The paper contains typos, like in the equation under line 138 where the term $x_t$ is never defined.

* On a methodological level, I find the contribution of the paper somehow marginal. Until Section 3.1, the authors present known, or easy-to-derive, results from the literature. The correspondence between the weighted loss and the ELBO is even explicitly stated in Kingma2021 (eq. (19) and the corresponding section)

* The derivation under line 173 is not rigorous for the same motivations mentioned above: the set of time indices is never properly defined.  Even the related proofs in Appendix C does not use a proper mathematical language.
While I understand the change of notation between eqs 12 and 13 ($z_t \to z_\lambda$) the presentation needs improvements.

* Finally the experimental validation is limited in several aspects. First, for a paper whose main contribution is on the exploration of different weighting schemes, the authors only experiment on a single dataset (ImageNet 64x64). Is this sufficient to draw any general conclusion about such aspect?

* None of the proposed weighting schemes achieve the performance of the existing baseline (EDM, Karras 2022) which has a FID score of 1.36. In general, the improvements are marginal, since we do not have access to the source code it is not possible to verify whether differences between considered schemes and baselines are due to the weighting schemes or other implementation details.


**Questions:**

Given the observations in the "weaknesses" section:

* Could you please emphasize the main contribution when compared to prior work such as Kingma2021 (eq. (19) and the corresponding section)?

* Can you clarify the doubts about mathematical details listed in the weaknesses section of the review?

* Can you comment on the generality of the experimental results, and how your conclusion could extend to other datasets?



==== Post rebuttal message, prior to reviewer discussion ====


Dear Authors, as a post rebuttal message, I am willing to increase my score to a 6: weak accept. I also have changed the scores to what concerns soundness, presentation and contribution.

I will eventually edit again my official review after the reviewer discussion phase, but for now I would like to thank you for your effort in addressing my concerns.

**Limitations:**

I couldn't find a limitation nor societal impact section, neither a short paragraph on these points.

---

> ### Author Rebuttal · Authors · 2023-08-08
>
> Thank you for your detailed review and valuable comments. Please note our top-level comment with additional experimental and theoretical results. Below we address specific questions.
>
> **Q: I couldn't find a limitation nor societal impact section, neither a short paragraph on these points.**
>
> A: Please find these sections in the PDF under "Supplementary Material" link, specifically Sections H and I. At the time of full-paper submission (one week before supplementary submission deadline), we accidentally appended an incomplete Appendix to the main paper. Please ignore that Appendix, and view the complete Appendix, which includes the relevant Sections, under the link of "Supplementary Material".
>
> **Q: Could you please emphasize the main contribution when compared to prior work such as [Kingma et al, 2021] (eq. (19)) and the corresponding section?**
>
> A: Our contributions are:
> 1. We explicitly derive the weighting functions and noise schedules for a large number of diffusion objectives used in the literature.
> 2. We show that the weighted loss is invariant to the noise schedule, except for its endpoints. This was only shown to hold for the unweighted loss (ELBO) by [Kingma et al, 2021].
> 3. We show that if the weighting function is monotonic, then the objective is equivalent to maximizing the ELBO combined with a form of DistAug, i.e., a form of data distribution augmentation. This improves our understanding of weighted diffusion training objectives, where the weighting is essential for best sample quality. We'd like to emphasize that this derivation (in Section 3.1 and the Appendix B and C) is non-trivial, hasn't been derived or stated in previous work, and potentially very valuable for our peers.
> 4. We experiment with new monotonic weightings, and demonstrate SOTA FID and Inception scores on high-resolution (128x128, 256x256 and 512x512) ImageNet generation. This demonstrates that we do not need to deviate from the ELBO objective to get good qualitative results, as long as we combine it with DistAug. We hope this could inspire future work along this direction.
>
> **Q: Can you clarify the doubts about mathematical details listed in the weaknesses section of the review?**
>
> A: Recall that for a discrete-time model, we have the forward process forming a conditional joint distribution $q(z_0, z_{\Delta t}, z_{2\Delta t}, ..., z_{T\Delta t}|x)$, and a generative model forming a joint distribution $p(z_0, z_{\Delta t}, z_{2\Delta t}, ..., z_{T\Delta t})$, assuming $T\Delta t= 1$. Like "Variational Diffusion Models" [Kingma et al, 2021] and much of the literature on ODEs and SDEs, we treat the continuous-time model as a discrete-time model, but taken in the limit where the number of timesteps $T$ between 0 and 1 goes to infinity. So, $dt = \lim_{0 \leftarrow \Delta t} \Delta t = \lim_{T \to \infty} 1/T$. This leads to the continuous-time forward process $q(z_{0,...,1}|x)$ and generative model $p(z_{0,...,1})$.
>
> Analogously, the KL divergence $D_{\text KL}(q(z_{0,...,1}|x)||p(z_{0,...,1}))$ is defined the same as the KL divergence of the discrete-time model:
> $$E_q\left[D_{\text KL}(q(z_1|x)||p(z_1)) + \sum_{i=1}^{T}D_{\text KL}(q(z_{(i-1)\Delta t}|z_{i\Delta t}, x))||p(z_{(i-1)\Delta t}|z_{i\Delta t}))\right]$$
> but taken in the limit where the number of timesteps $T$ between 0 and 1 goes to infinity, and time step size $\Delta t$ goes to $0$. This approach was also taken by [Kingma et al, 2021].
>
> For notational brevity we didn't include these limits in the notation for the proofs under line 173 and in our Appendix C. However, we can see why this caused unnecessary confusion. We've already improved the notational rigor of our manuscript, and will keep improving for the camera-ready version.
>
> **Q:[...] I find confusing and counterproductive to switch back and forth between different notations (for example dropping or not the dependency on $t$ of $\lambda$).**
>
> A: In Section 2.2 (line 99), we mentioned *"Below we sometimes denote $\lambda$ as $\lambda_t$ to emphasize that it is a bijective function of $t$."* We agree that it is not ideal to switch between these notations. For the final version we will put more effort in finding a way to minimize the need to use different notations.
>
> **Q: Can you comment on the generality of the experimental results, and how your conclusion could extend to other datasets?**
>
> A: As noted in the top-level comment, we expanded our experiments to high-resolution (256x256 and 512x512) ImageNet generation, and found that weighting functions that performed well in earlier lower-resolution experiments also perform well in these high-resolution settings. In our initial draft, we also have results on ImageNet 128x128 generation (Section 6.2), besides ImageNet 64x64. For high-resolution ImageNet generation (128x128, 256x256, 512x512), we achieved SOTA FID and Inception scores among approaches without guidance, and demonstrated significant improvements compared to the baseline approach Simple Diffusion [Hoogeboom et al, 2023] that we built upon. Please check the top-level comment for more details.
>
> Given these results, it's plausible that the tested weighting functions will also work well on other datasets of images.
>
> It's hard to say a priori which weighting functions would work well for other modalities in practice, like audio. However, in our newest version of our manuscript, we show that VoiceBox [Le et al., 2023], a state-of-the-art model for audio, uses a training objective that is equivalent to the weighted loss with an exponential weighting $w(-\lambda/2)$. In experiments we found that this weighting works well for images. Therefore, we now know that this weighting function works well for both images and audio.
>
> On the other hand, all our theoretical results are agnostic to the types of data used, thus our theoretical conclusions naturally extend to other modalities.
>
> **Q: Typo under line 138.**
>
> A: We have fixed it in our latest draft. Thanks for the careful reading!

---

> > ### Comment · Reviewer_Qo9S · 2023-08-17
> > **Thanks for the careful rebuttal to the reviews**
> >
> > Dear authors,
> > thanks a lot for the effort you put in your rebuttal, for all reviews.
> >
> > On the theoretical side, concerning my comments on rigor and notation, you addressed all the points and clarified your notation, as well as the continuous-time limit formulation. Thank you.
> > I think you managed to identify more clearly what sets apart the submitted paper from [Kingma et al, 2021], namely to extend their results to more general losses. I see now this to be interesting, with some benefits for practitioners who seek high sample quality.
> >
> > On the empirical side, I have checked the rebuttal pdf with the new results on higher resolution ImageNet data. These new results illustrate some cases in which the proposed approach does indeed surpass or is well on par with SOTA. Also the general comment to explain why in one experimental setting you didn't obtain the best FID is well detailed and convincing.
> > I am still convinced that this work could have included at least one different dataset, even an easier one such as CelebA-HQ or similar, to consolidate our understanding of the generality of the proposed technique.
> >
> > Overall, I have noticed concrete improvements to the draft, and I am going to raise my score in the official review.
> >
> > Again, thank you very much for your work.

---

> > > ### Author Response · Authors · 2023-08-17
> > >
> > > Dear reviewer,
> > >
> > > Thank you for raising your score to a 6 (Weak accept) and for explaining why.
> > >
> > > Could you perhaps elaborate on what your remaining criticism is, i.e. why you did not give a higher rating? Is there something we could do to further improve your rating?
> > >
> > > Again, thank you for raising your rating.

---

> ### Author Response · Authors · 2023-08-14
>
> Dear Reviewer,
>
> Thank you again for reviewing our manuscript. We have tried our best to address your questions (see our rebuttal in the top-level comment and above), and revised our paper by following suggestions from all reviewers.
>
> Please kindly let us know if you have any follow-up questions or areas needing further clarification. Your insights are valuable to us, and we stand ready to provide any additional information that could be helpful.

---

### Official Review · Reviewer_bjDS · 2023-07-06

**Soundness:** 3 good
**Presentation:** 4 excellent
**Contribution:** 4 excellent
**Rating:** 8
**Confidence:** 3

**Summary:**

The paper takes a systematic approach to address two important aspects of the implementation of diffusion-based generative models: the objective function and the noise schedule, both of which have been conventionally set in a relatively empirical fashion ($i.e.$, based on sample quality).

The paper first frames the noise schedule in terms of the log SNR $\lambda_t$ as a function of time $t$, then writes the diffusion model objective in the form of a weighted denoising score matching objective, Eq. (6). By doing so, the paper unifies the existing works as special cases under a generic form of the objective function with its corresponding noise schedule.

The paper continues with a theoretical analysis of the diffusion model objective, by first revealing the ELBO objective as one particular realization of the weighted objective, then provides the key theoretical result via Theorem 1, which states that **under the assumption of a monotonically increasing weighting function $w(\lambda_t)$, the generic diffusion model objective Eq. (6) is equivalent to a negative ELBO objective of noise-perturbed data (a form of data augmentation, controlled by the noise schedule $\lambda_t$)**. This result has recast the ELBO as the class of objective functions for training diffusion models that are capable of producing high-quality samples.

With the theoretical results, the paper proposes several new monotonic weighting functions, and demonstrates through experiments that monotonic weighting enables the trained diffusion models to generate high quality samples (competitive with the best diffusion models) for image tasks in terms of FID and IS, including a new SOTA FID score of $1.75$ on class-conditional $128\times 128$ ImageNet dataset.

**Strengths:**

* Among all degrees of freedom for the implementation of a diffusion-based generative model, setting the weights at each noise level for the objective function, and noise scheduling, are two crucial aspects for its success that have traditionally been mainly based on empirical choices. By addressing these two aspects with a systematic and rigorous approach, this work fills an important gap for the diffusion model research community, thus is a well-motivated and potentially very beneficial work.
* By recasting the diffusion model objective under the restriction of the weighting function as an ELBO objective, the work can bring back this important class of objective functions along with the existing literature on training with an ELBO objective.
* The paper proposes several new monotonic weighting functions for the weighted diffusion objective.
* The paper is overall well-organized and clearly written: it can be viewed as a good reference that introduces the diffusion-based generative modeling framework collectively; the color-coding for different components of the objective functions is a good design for presenting derivations.

**Weaknesses:**

* The part of the paper that derives the main conclusion of the paper (the diffusion objective is equivalent to the ELBO with data augmentation), specifically Line 181-185, went a little bit too fast-paced and might lack of some details to facilitate understanding; some descriptions don’t exactly match the mathematical expressions:
    * The KL divergence $D_{KL}(q(z_{t,\dots,1}\vert x)\vert\vert p(z_{t,\dots,1}))$ being “the negative ELBO of noise-perturbed data, plus a constant” is not immediately clear. It perhaps would be nice for the authors to provide the derivations, to establish $D_{KL}(q(z_{t,\\dots,1}| x)|\vert p(z_{t,\\dots,1})) =-\\mathbb{E}_{q(z_t|x)}[\\text{ELBO of }\\log p(z_t)]-\\mathcal{H}(q(z_t|x))$, where

   	 $\text{ELBO of }\\log p(z_t)=\\mathbb{E}_{q(y | z_t, x)}[\\log p(z_t, y)-\\log q(y | z_t, x)],$ in which $y=z\_{t+\\triangle t,\dots,1},$ and $\\triangle t$ is the time increment.

    	Similarly, the inequality in Eq. (9), $D_{KL}(q(z_{t,\\dots,1}| x)|\vert p(z_{t,\\dots,1})) = D\_{KL}(q(z_t | x)|\vert p(z_t)) + \\mathbb{E}_{q(z_t | x)}\big[D\_{KL} (q(y | z_t,x)||p(y | z_t))\big]\\ \geq D\_{KL}(q(z_t | x)|\vert p(z_t))$.

    	It might be helpful to clarify these terms, since Line 181-182 seems to be the very part that constructs the main claim of the paper $s.t.$ the diffusion objective can be viewed as the (negative) ELBO of noise-perturbed data.
    * Line 184-185 says that minimizing $\mathcal{L}\_w(\\theta)$ with a monotonic weighting “is equivalent to maximizing the ELBO of noise-perturbed data”. The expression is a bit strange as it indicates there is only one ELBO term to be optimized, while the objective function is shown in this section to be viewed as a collection of different ELBO terms. Therefore, this simplified description might cause some misinterpretation.
    * The underbracket text for Eq. (9) and (10) are both “Neg. log-likelihood of noise-perturbed data”, while the expressions are different expectations of $-\\log p(z_t)$.
* The authors mentioned that re-establishing the diffusion objective as an ELBO “allows for a direct apples-to-apples comparison … with other likelihood-based models” (Line 63-64), but it’s unclear on how such comparison can be made:
    * Does it refer to the training loss when the autoregressive transformer is trained *with the same data augmentation scheme*? If so, it would only make sense if the adaptation of the data augmentation scheme would not worsen the autoregressive transformer’s performance in the first place.
    * Or does it refer to the computation and comparison of the exact likelihood?

    It would be nice for the authors to elaborate more on how such comparisons can be made, as it occurs noticeably at the end of both the Introduction and the Conclusion section.

> Minor Issues
* Line 10 in the Abstract says that the weighted loss “equals the ELBO”. Based on Section 3.1, would it be more accurately described as “equivalent to optimizing the ELBO”?
* Line 39 “This has has led”.
* Line 42 “training objective” might be “training objectives”.
* Section 2.5 has both forms for the score network,  “$s_{\theta}(z;\lambda)$” and “$s_{\theta}(z,\lambda)$”.
* Line 142 comma between “$x$-prediction” and “$v$-prediction”.
* Eq. (8) 2nd and 3rd line: the expectation has “$t\sim U(0,1))$”.
* Table 5 text box ends with two periods.
* Line 62 & 295 the expression “The newfound equivalence between monotonic weighting and the ELBO with data augmentation” seems odd; do the authors mean “The newfound equivalence between monotonically weighted diffusion objective and the ELBO with data augmentation”?
* It’s a bit confusing to first define $\lambda$ as the log SNR (Line 86), then call it the noise level (Line 91): the sentence in Line 93-94, “At timestep t = 0 the noise level … $\lambda$ is high, meaning very little noise; at t = 1 the noise level $\lambda$ is low, meaning a lot of noise”, sounds a bit odd.
* The term “noise schedule” is overloaded: in the paper, it simultaneously means the log SNR term $\lambda_t$ itself (Line 149), the function $f_{\lambda}$ (Line 90), and the distribution $p(\lambda)$ (Line 510). It’s clear to me that each “noise schedule” would result in a particular value of the variance ${\sigma_{\lambda}}^2$ at each timestep, but I’m not sure if such overloading would create any issue in the future, thusI’ll just point it out here.
* Line 179-180 the weighting function $w$ is simultaneously a function of $\lambda_t$ and of $t$.

**Questions:**

* Could the authors provide some explanation for the following sentences:
    * Line 108-109, “During sampling, $p(\lambda)$ provides the relative amount of time the sampler spends at different noise levels.” It’s not immediately clear what it means, as it seems that by sampling $t\sim U(0,1)$, each noise level has an equal chance of being sampled.
    * Line 179-180, “The probability distribution $p_w(t)$ has Dirac delta peak of typically very small mass $w(0)$ at $t = 0$.” Does this mean that the almost noiseless sample contributes very little to the objective computation?
* Line 262-265: have the authors run experiments for a $v$-prediction model combined with a non-monotonic weighting function?
* In Figure 1 Right, only the “EDM sampling” noise schedule has $p(\lambda)$ monotonically decreasing as a function of $\lambda$. Is there any interpretation of that, $e.g.$, advantages for the EDM sampling scheme to monotonically decrease?
* It’s a bit confusing on why the noise schedule is casted as a probability distribution $p(\lambda_t)$ (besides the importance sampling interpretation): $e.g.$, the cosine schedule has the variance at each timestep as a deterministic function of time. Why does $\lambda$ need to be sampled – is it because timesteps need to be sampled during training?

**Limitations:**

The authors have discussed both limitations and potential negative social impact via Appendix H and I from Supplementary Material, which speak for the community well.

---

> ### Author Rebuttal · Authors · 2023-08-08
>
> Thank you for your detailed review and thoughtful comments. Please note our top-level comment with additional experimental and theoretical results. Below we address specific questions.
>
> **Q: The part of the paper that derives the main conclusion of the paper [...], went a little bit too fast-paced [...].**
>
> A: Thank you for your valuable feedback. We have expanded and clarified this section following your suggestion, and added a detailed derivation to the Appendix.
>
> **Q: The authors mentioned that re-establishing the diffusion objective as an ELBO “allows for a direct apples-to-apples comparison … with other likelihood-based models” (Line 63-64), but it’s unclear on how such comparison can be made. [...].**
>
> A: We will clarify this part of our manuscript. As we mentioned in Section 5 (Related work), the weighted loss in Equation 10 is a type of DistAug [Jun et al., 2020], a powerful method of training data distribution augmentation for generative models. In DistAug, the generative model is conditioned on the data augmentation parameter while training, but conditioned on "no augmentation" during sampling. It was shown by [Jun et al., 2020] that DistAug can substantially improve sample quality of autoregressive models on images.
>
> In our case the augmentation is simple additive noise with noise level $\lambda_t$, and the model $p(z_t)$ as provided by the score network is conditioned on the noise level $\lambda_t$. After training, the sampling procedure provides samples from $p(z_t)$ at $t=0$, which has virtually no augmentation. So this is directly analogous to DistAug where the model is trained with various types of augmentation (and conditioned on the augmentation indicator), and sampled by setting the augmentation indicator to "no augmentation".
>
> So, we can optimize other types of models similarly using DistAug, where we train with a distribution $t \sim p_w(t)$, translating to a distribution over levels $\lambda_t$, where the generative model is conditioned on the augmentation indicator $\lambda_t$. After training, we sample from the model with the augmentation indicator set to "no augmentation", i.e. $t=0$.
>
> **Q: About minor issues.**
>
> A: All points raised here are great. We appreciate your careful reading of our draft, and we will incorporate the suggestions into our updated draft.
>
> **Q: Line 108-109, "[...], $p(\lambda)$ provides the relative amount of time the sampler spends at different noise levels". It’s not immediately clear what it means, [...].**
>
> A: Since the noise level $\lambda$ is a nonlinear function of time $t$, sampling $t$ uniformly results in a non-uniform distribution of noise levels $\lambda$. During sampling, $p(\lambda)$ provides the density of sampling steps at noise level $\lambda$.
>
> **Q: Line 179-180, “The probability distribution $p_w(t)$ has Dirac delta peak of typically very small mass $w(0)$ at $t=0$.” Does this mean that the almost noiseless sample contributes very little to the objective computation?"**
>
> A: Yes, that's correct. There's typically still somewhat non-trivial total probability mass over all low noise levels, but the weight over these noise levels is relatively low, since it corresponds to modeling fine pixel details, which are of relatively lower importance for the quality of samples than the more abstract image information, which is modeled at the higher noise levels. Of all the plotted weighting functions, only the ELBO weighting gives high weight to low noise levels, but it also leads to the worst FID scores.
>
> **Q: Line 262-265: have the authors run experiments for a $v$-prediction model combined with a non-monotonic weighting function?**
>
> A: We have not previously. For $v$-parametrized model, we used the $v$-prediction loss $||{\textbf v} - \hat{{\textbf v}}||^2_2$ widely adopted in the literature as the baseline, which corresponds to a monotonic weighting function $\exp(\lambda / 2)$. We are running an experiment using $v$-parametrized model and $\epsilon$-prediction loss ($L_{\text simple}$), resulting in a non-monotonic weighting function ${\text sech}(\lambda /2)$. We will report the result later.
>
> **Q: In Figure 1 Right, only the “EDM sampling” noise schedule $p(\lambda)$ has monotonically decreasing as a function of $\lambda$. Is there any interpretation of that, [...]?**
>
> A: We did notice that the "EDM sampling" schedule spends a lot more time at the very high and the very high noise levels, and less time at the intermediate noise levels. The reason might be that the optimal design of training and sampling noise schedules can be very different: for training, the optimal schedule seeks to minimize the variance of the training objective, while for evaluation, the optimal design should minimize the "truncation error" accumulated along the discretized reverse-ODE or reverse-SDE trajectory, which is also the intuition of EDM sampling schedule. As noted in Appendix D.1 of [Karras et al, 2022], in practice, the truncation error of Euler's method (1st order ODE solver) of the reverse-ODE is monotonically decreasing over $\lambda$. Therefore, we should allocate more sampling steps in low $\lambda$ region, to allow for using small step sizes with smaller truncation error. This leads to the monotonically decreasing $p(\lambda)$.
>
> **Q: It’s a bit confusing on why the noise schedule $p(\lambda)$ is casted as a probability distribution (besides the importance sampling interpretation); [...]. Why does $p(\lambda)$ need to be sampled – is it because timesteps need to be sampled during training?**
>
> A: Yes, since time $t$ is sampled randomly during training, the noise level $\lambda$ has a distribution $p(\lambda)$ during training. We think the importance sampling interpretation is important, since it allows us to intuitively understand that given a fixed weighting function $w(\lambda)$, the loss is invariant to the choice of noise schedule $p(\lambda)$; $p(\lambda)$ only affects the variance of the Monte Carlo estimator of the loss.

---

> ### Author Response · Authors · 2023-08-14
>
> Dear Reviewer,
>
> Thank you again for reviewing our manuscript. We have tried our best to address your questions (see our rebuttal in the top-level comment and above), and revised our paper by following suggestions from all reviewers.
>
> Please kindly let us know if you have any follow-up questions or areas needing further clarification. Your insights are valuable to us, and we stand ready to provide any additional information that could be helpful.

---

> > ### Comment · Reviewer_bjDS · 2023-08-17
> > **Response to Author Rebuttal**
> >
> > Dear authors,
> >
> > Thank you very much for your thorough response to my comments and questions, it has helped me to gain a better understanding of the paper and has clarified my questions. I plan to keep my current score.

---

### Official Review · Reviewer_eoU3 · 2023-07-06

**Soundness:** 4 excellent
**Presentation:** 4 excellent
**Contribution:** 4 excellent
**Rating:** 8
**Confidence:** 3

**Summary:**

This paper presents new theoretical insights on denoising diffusion objective. Namely, training a diffusion model with a non-uniform weighting can be expressed as a weighted integral of ELBOs, and monotonic weighting results in training ELBO with additive Gaussian noise augmentation. The authors also propose an adaptive noise schedule to ensure all losses are trained uniformly over time and show the adaptive schedule with a monotonic weighting attains SOTA FID on ImageNet.

**Strengths:**

Although not shown in this work, the interpretation that denoising diffusion with monotonic weighting trains ELBO with data augmentation means that the same data augmentation can be applied to other likelihood-based models for a fair comparison to diffusion models. And, we would have a clearer understanding of sample quality and likelihood of various model classes.

**Weaknesses:**

-

**Questions:**

-

**Limitations:**

Yes, the authors addressed limitations, potential negative societal impact, and mitigation.

---

> ### Author Rebuttal · Authors · 2023-08-08
>
> Thank you for your constructive comments. We strongly agree with you that understanding weighted diffusion training objectives as ELBO with data augmentation gives us a way to fairly compare diffusion models with other likelihood-based models, and indicates that ELBO objective is still compatible with high sample quality. Please note our top-level comment with additional experimental and theoretical results.

---

> ### Author Response · Authors · 2023-08-14
>
> Dear Reviewer,
>
> Thank you again for reviewing our manuscript. We have tried our best to address your questions (see our rebuttal in the top-level comment and above), and revised our paper by following suggestions from all reviewers.
>
> Please kindly let us know if you have any follow-up questions or areas needing further clarification. Your insights are valuable to us, and we stand ready to provide any additional information that could be helpful.

---

> ### Comment · Area_Chair_3zyb · 2023-08-20
> **Reminder from AC**
>
> Dear reviewer,
>
> The author-reviewer discussion period ends in 2 days. Please review the authors' rebuttal and engage with them if you have additional questions or feedback. Your input during the discussion period is valued and helps improve the paper.
>
> Thanks, Area Chair

---

### Official Review · Reviewer_DESf · 2023-07-14

**Soundness:** 4 excellent
**Presentation:** 4 excellent
**Contribution:** 4 excellent
**Rating:** 9
**Confidence:** 4

**Summary:**

This is a nice paper that provides a comprensive framework for diffusion objectives, giving the ELBO the driver's seat.

The paper shows that all of the axes along which diffusion objectives differ (elbo vs Ho-perceptual weighting, noise schedules, different parameterizations like x vs epsilon vs v) can all be seen under a type of weighted and importance sampled ELBO. Moreover, the paper show that the weightings that don't exactly look like the original ELBO can be made equal to it under the right data augmentation. Finally, the authors also present a few techniques for adaptive/learnable importance sampling distributions to help assemble all the aforementioned ingredients in a computationally efficient manner. Finally, the authors show conveying results for image quality for ELBO trained models, whereas people usually result to non-ELBO trained models to do this.

**Strengths:**

I'm a big fan of papers like this since their primary aim to build solid understanding for people that already work with diffusions but that may have some doubts about how the common moving parts relate to each other (noise schedules, elbo weighting, parameterizations, etc).

On top of the above, they also present great results for ELBO trained models, whereas there is some common beliefs that non-elbo "perceptual weighting" (what the authors call "reweighted losses") are the only ones that can be used for good image quality.

**Weaknesses:**

No comments/weaknesses.

The adaptive importance sampling also comes up in the following work:

Improved Techniques for Maximum Likelihood Estimation for Diffusion ODEs https://arxiv.org/abs/2305.03935

This is completely fine since:
- I can see how both works would arrive at this algorithmic detail independently, given that both dive deeply into related but distinct variance issues in diffusion elbos.
- both works present a lot of additional insights and techinques beyond this

I do find it appropriate to cite and discuss relationship in either case though, including possible similarities and differences, since it will really help readers of both papers.

**Questions:**

Just a brief suggestion:

- around eq 3 it would be good to mention that the f(x,t)-g^2(t) s_\theta(x,t) form is a backward time SDE since some other authors use the g^2(T-t)s_\theta(x,T-t) -f(x, T-t) forward time SDE.

- Not necessary to address but I'm quite curious which of the invariance and change of variables bits apply for multivariate (augmented) diffusions like CLD (https://arxiv.org/abs/2112.07068), MDMs (https://arxiv.org/abs/2302.07261), PSLD (https://arxiv.org/abs/2303.01748), and FP-Diffusion (https://arxiv.org/abs/2206.10365). You can't quite take the g^2/sigma^2 weighting out of the squared error like in the scalar diffusion case, but I'm sure there should be some relevant version of your results.

**Limitations:**

Yes.

---

> ### Author Rebuttal · Authors · 2023-08-08
>
> Thank you for your valuable comments and kind words to our work. Please note our top-level comment with additional experimental and theoretical results. Below we address specific questions.
>
> **Q: Cite and discuss relationship with “Maximum Likelihood Estimation for Diffusion ODEs” (https://arxiv.org/abs/2305.03935).**
>
> A: We appreciate your pointer to this related work, which also proposes noise schedules with the purpose of reducing the variance of diffusion objectives. They proposed two approaches: 1) design $p(\lambda)$ such that the coefficient of the mean squared error of normalized v-parametrization is a time-invariant constant; 2) learning a noise schedule network $\lambda(t)$ by minimizing the variance of the training objective, similar to [Kingma et al, 2021]. While 2) in theory can seek the optimal importance sampling, it introduces extra computational cost with additional gradient operations. We therefore choose a less expensive approach: we maintain the exponential moving average of $w(\lambda)||\epsilon - \hat{\epsilon}_\theta||^2_2$, and adaptively update the noise schedule $p(\lambda)$ to be proportional to that at each iteration. Similar approach was proposed by [Nichol et al, 2021].
>
> We will cite this paper and discuss the relationship in our final version, as you suggested.
>
> **Q: Not necessary to address but I'm quite curious which of the invariance and change of variables bits apply for multivariate (augmented) diffusions.**
>
> A: That's a very good point! A more general formulation would be interesting, which we leave open for future work.
>
> Thanks for your suggestion on clarifying Eq 3, which we have incorporated to the updated draft.

---

> ### Author Response · Authors · 2023-08-14
>
> Dear Reviewer,
>
> Thank you again for reviewing our manuscript. We have tried our best to address your questions (see our rebuttal in the top-level comment and above), and revised our paper by following suggestions from all reviewers.
>
> Please kindly let us know if you have any follow-up questions or areas needing further clarification. Your insights are valuable to us, and we stand ready to provide any additional information that could be helpful.

---

### Official Review · Reviewer_kL4L · 2023-08-01

**Soundness:** 3 good
**Presentation:** 3 good
**Contribution:** 3 good
**Rating:** 6
**Confidence:** 2

**Summary:**

The paper tackles the loss function of diffusion models from a theoretical point of view. The paper studies the relationship between the weighted loss and the ELBO objective. The paper shows that the diffusion model objectives can be formulated as a special case of weighted loss. In addition to the theoretical contribution, the authors propose monotonic weightings and provide an empirical study of monotonic and non-monotonic weighting.

**Strengths:**

- The problem studied is important
- Formulating the different diffusion model loss as weighted loss is interesting
- The paper presented a detailed theoretical analysis
- Experimental results support the proposed method

**Weaknesses:**

- Adding more experimental results could have been helpful but this is not a major point.

**Questions:**

NA

---

> ### Author Rebuttal · Authors · 2023-08-08
>
> Thank you for your thoughtful feedback. Please note our top-level comment with additional experimental and theoretical results. Below we address specific questions.
>
> **Q: Adding more experimental results could have been helpful.**
>
> A: Please see our top-level comment and the PDF attached to it. We have added high-resolution ImageNet generation results (i.e. 256x256 and 512x512) to further demonstrate the effectiveness our approach. With our 'EDM-monotonic' function and adaptive noise schedule, we achieved SOTA FID and Inception scores on high resolution ImageNet (i.e., all three resolutions of 128, 256, 512).

---

> ### Author Response · Authors · 2023-08-14
>
> Dear Reviewer,
>
> Thank you again for reviewing our manuscript. We have tried our best to address your questions (see our rebuttal in the top-level comment and above), and revised our paper by following suggestions from all reviewers.
>
> Please kindly let us know if you have any follow-up questions or areas needing further clarification. Your insights are valuable to us, and we stand ready to provide any additional information that could be helpful.

---

> ### Comment · Area_Chair_3zyb · 2023-08-20
> **Reminder from AC**
>
> Dear reviewer,
>
> The author-reviewer discussion period ends in 2 days. Please review the authors' rebuttal and engage with them if you have additional questions or feedback. Your input during the discussion period is valued and helps improve the paper.
>
> Thanks, Area Chair

---

### Author Rebuttal · Authors · 2023-08-08

Dear Reviewers,

**Please see the attached PDF for a one-page PDF with a summary of added experimental results.**

We would like to thank all reviewers for providing constructive feedback that helped us improved the paper. We are encouraged that reviews think our paper:
- "provides a comprensive framework for diffusion objectives" (Reviewer DESf),
- "fills an important gap for the diffusion model research community" (Reviewer bjDS),
- provides "a clearer understanding of sample quality and likelihood of various model classes" (Reviewer eoU3) and was well-supported by both "detailed theoretical analysis" and "experimental results" (Reviewer kL4L).

We have been working diligently on improving the paper on several fronts, addressing your critique. Below, we summarize the changes that we have made in an updated draft.

**1. High-resolution (256x256 and 512x512) ImageNet results**

We provide new experimental results, with state-of-the-art FID and Inception scores for high-resolution ImageNet generation (256x256 and 512x512). Please see the attached PDF for a table of the summary of quantitative evaluations and a figure for generated samples by our approach.

With the shifted version of ‘EDM-monotonic’ weighting, we achieved state-of-the-art FID and Inception scores on all three resolutions (i.e., 128x128, 256x256, 512x512) of ImageNet generation among all approaches without any guidance. With classifier-free guidance (CFG) [Ho and Salimans, 2022], our method outperforms all diffusion-based approaches on resolutions 128 and 512. On resolution 256, our method only falls a bit behind Gao et al. [2023] and Hang et al. [2023], both of which were build upon the latent space of a pretrained auto-encoder from Stable Diffusion [Rombach et al., 2022a], that was trained on a much larger image dataset than ImageNet, while our model was trained on the ImageNet dataset only. It is worth noting that we achieve significant improvements on the Simple Diffusion baseline model [Hoogeboom et al., 2023] that we built upon, on all resolutions, with and without guidance. In principle it is possible to apply our proposed weighting functions and adaptive noise schedules to other diffusion-based approaches such as Gao et al. [2023], possibly further improve their performance.

We will provide additional samples from our trained models in a new section in the updated Appendix.

**2. More weighting functions**

We expanded Table 1 and the Appendix with more weighting functions used in the literature. Specifically, we added the weighting function corresponding to Flow Matching with Optimal Transport (FM-OT) [Lipman et al., 2022]: $w(\lambda) = \exp(-\lambda/2)$, and the weighting function corresponding to InDI [Delbracio and Milanfar, 2023]: $w(\lambda) = \exp(−\lambda)\text{sech}^2(\lambda/4)$. The detailed derivations will be provided in the Appendix. Interestingly, both weighting functions are monotonic.

**3. More helpful visualizations and analysis**

- We added an analysis on the relationship with low-bit training to the Appendix.
- We added a visualization of integration-by-parts, used in our main proof, to the Appendix.

Please see our reviewer-specific feedback for more information.

---

### Decision · Program_Chairs · 2023-09-21

**Decision:**

Accept (oral)

**Comment:**

After a thorough evaluation by five reviewers and considering the authors' detailed rebuttal, I recommend acceptance of this paper as an oral presentation at NeurIPS 2023.

### Summary
The paper presents a comprehensive theoretical analysis of diffusion models, elucidating the relationships between the weighted loss and the Evidence Lower Bound (ELBO) objective. By investigating monotonic weighting and introducing new monotonic weightings, the authors establish connections between the weighted loss and ELBO with data augmentation. They demonstrate the effectiveness of their approach through experiments on ImageNet, achieving competitive results.

### Strengths
Important Contribution: The work fills a significant gap in the diffusion model research community and provides valuable insights. Reviewer DESf praised it, stating, "I'm a big fan of papers like this since their primary aim [is] to build solid understanding [...] and they also present great results for ELBO trained models."

Theoretical Clarity: The theoretical derivation and insights were appreciated by most reviewers. Reviewer eoU3 noted the contribution as "excellent."

Experimentation: The empirical results were seen positively by several reviewers, and the additional high-resolution ImageNet results presented in the rebuttal further strengthened this aspect.

### Concerns
Mathematical Rigor: Concerns were raised about mathematical rigor and notation, particularly by Reviewer Qo9S. However, the authors provided clarifications in their rebuttal, including insights on their use of continuous-time limits and notation.

Experimental Diversity: Reviewer Qo9S expressed concerns about the lack of diversity in datasets, but the authors provided additional experimental results in their rebuttal and explained the relevance to other modalities.

### Conclusion

Considering the strong theoretical contributions, the efforts to clarify mathematical details, and the effective experimental validation, I believe this paper would make a valuable addition to the conference as an oral presentation. The work builds on existing research in a meaningful way, providing practical insights that could influence the design and understanding of diffusion models. Reviewer DESf's comment encapsulates the sentiment well: "This is a nice paper that provides a comprehensive framework for diffusion objectives, giving the ELBO the driver's seat." The thoughtful rebuttal and additional empirical support further solidify the case for acceptance.